# How to Design Stable Machine Learned Solvers For Scalar Hyperbolic PDEs

## Abstract

Machine learned partial differential equation (PDE) solvers trade the robustness
of classical numerical methods for potential gains in accuracy and/or speed. A key
challenge for machine learned PDE solvers is to maintain physical constraints that
will improve robustness while still retaining the flexibility that allows these methods
to be accurate. In this paper, we show how to design solvers for scalar hyperbolic
PDEs that are stable by construction. We call our technique 'global stabilization.'
Unlike classical numerical methods, which guarantee stability by putting local
constraints on the solver, global stabilization adjusts the time-derivative of the
discrete solution to ensure that global invariants and stability conditions are satis-
fied. Although global stabilization can be used to ensure the stability of any scalar
hyperbolic PDE solver that uses method of lines, it is designed for machine learned
solvers. Global stabilization's unique design choices allow it to guarantee stability
without degrading the accuracy of an already-accurate machine learned solver.

## 1   Introduction

Scientists and engineers are interested in solving partial differential equations (PDEs). Many PDEs
cannot be solved analytically, and must be approximated using discrete numerical algorithms. We
refer to these discrete numerical algorithms as PDE solvers. The fundamental challenge for PDE
solvers is to balance between two competing objectives: first, to find an accurate approximation to
the solution of the equation, and second, to do so with as few computational resources as possible.

In recent years, scientists and engineers have attempted to use machine learning (ML) to design
new and better PDE solvers [41, 2, 45, 31, 16, 44, 3, 42]. On certain problems, machine learned
PDE solvers have achieved high accuracy at low computational cost [22, 39, 24, 13, 26]. However,
these high-performing machine learned PDE solvers suffer from at least two major problems. First,
they struggle to generalize to conditions outside of the training data. Second, they tend to have no
guarantees of stability and as a result the solution sometimes blows up as $t \to \infty$. For examples of this
second problem, see fig. 3a of [2] and fig. 9a of [45]. Consequently, [2] and [45] write that "figuring
out how to guarantee stability" of machine learned PDE solvers is an "important topic for future work."

We consider scalar hyperbolic PDEs written in conservation form, given by

$$\frac{\partial u}{\partial t} + \boldsymbol{\nabla} \cdot \boldsymbol{f}(u) = 0. \tag{1}$$

For an introduction to the mathematical properties of, classical numerical methods for solving, and
motivation for studying eq. (1), see [30]. If machine learned solvers for eq. (1) were somehow
perfectly accurate, then stability (see section 2) would not be a concern because the solver would
simply give the correct answer for all $t$. But, for a variety of reasons, machine learned solvers are not
and will never be perfectly accurate. Some amount of error is inevitable, so the question becomes:

Submitted to 36th Conference on Neural Information Processing Systems (NeurIPS 2022). Do not distribute.

how can we constrain the machine learned solver to give us the sorts of errors that we are willing to tolerate? Although the answer to this question is problem dependent, we take the view that with machine learned numerical methods, as with well-designed classical numerical methods, the solution should be guaranteed not to blow up as $t \to \infty$ (see section 3).

The purpose of this paper is to demonstrate how to design machine learned solvers for eq. (1) that ensure stability (see sections 4 and 5) without degrading the accuracy of the solution. These solvers guarantee both mass conservation and stability as $t \to \infty$ for a subset of PDEs that are highly relevant in the physical sciences and engineering. We call our technique 'global stabilization.' In particular, the global stabilization technique can be used as a 'hard' constraint on the model architecture of so-called 'hybrid' machine learned solvers (see section 6). We present the global stabilization technique in 1D and 2D for rectangular uniform grids with periodic boundary conditions (BCs). We note that the method can also be used when the right hand side (RHS) of eq. (1) is nonzero (see appendix A) and for non-periodic BCs and non-uniform grid spacing (see appendix B).

## 2 Stability of Scalar Hyperbolic PDEs

**Conservation properties:** eq. (1) implies that the scalar $\int u(\boldsymbol{x}, t) \, d\boldsymbol{x}$ is time-invariant, which we call 'conservation of mass.' In a 1D periodic system with $x \in [0, L]$, an integral over $x$ makes the invariance apparent: $\frac{d}{dt} \int_0^L u(x, t) \, dx = \int_0^L \frac{\partial u}{\partial t} \, dx = -\int_0^L \frac{\partial f}{\partial x} \, dx = f(0) - f(L) = 0$. In words: the total rate of change of $u$ is equal to the flux through the boundaries; for a periodic system this equals zero.

**Stability properties:** we begin with the entropy inequality [30, 38] given by

$$\frac{\partial S(u)}{\partial t} + \boldsymbol{\nabla} \cdot \boldsymbol{F}(u) \geq 0. \tag{2}$$

Equation (2) is satisfied for any concave entropy function $S(u)$, so long as the entropy flux $\boldsymbol{F}$ is defined as $\boldsymbol{F}(u) := \int^u S'(u) \boldsymbol{f}'(u) \, du$. Integrating eq. (2) over $x$ for a 1D periodic system where $x \in [0, L]$ shows that the total entropy is non-decreasing: $\frac{d}{dt} \int S(u) \, dx \geq F(0) - F(L) = 0$. By choosing $S(u) = -||u||_p$, where $||u||_p$ is defined as the $\ell_p$-norm $||u||_p := (\int |u|^p \, dx)^{1/p}$ for $1 \leq p < \infty$, we have the first stability property of eq. (1), which is that the $\ell_p$-norm of $u$ is non-increasing: $\frac{d}{dt} ||u||_p \leq 0$ for $1 \leq p < \infty$. Taking the limit as $p \to \infty$ gives a second stability property, which is that the $\ell_\infty$-norm of $u$ is non-increasing: $\frac{d}{dt} ||u||_\infty \leq 0$. There is a third stability property, called the 'total variation diminishing' (TVD) property, which is derived in [30]. For continuous $u$, the TVD property is that $\frac{d}{dt} \int_0^L \left| \frac{\partial u}{\partial x} \right| \, dx \leq 0$.

## 3 Stability of Discrete Numerical Methods for Scalar Hyperbolic PDEs

[30] writes that "the central philosophy of numerical analysis is to devise numerical schemes that preserve stability properties of the underlying continuous problem." We now review how classical techniques preserve stability properties.

The standard approach of solving time-dependent PDEs is to discretize the PDE in space, which generates a system of ordinary differential equations (ODEs), then to integrate those ODEs in time. This approach is called method of lines (MOL). A very common approach for solving conservation-form PDEs is by using some type of finite-volume (FV) method. FV methods divide the spatial domain into a number of cells, then use a scalar value to represent the solution average within each cell. For example, on the 1D domain $x \in [0, L]$ with uniform cell width, a FV method divides the domain into $N$ cells of width $\Delta x = L/N$ where the left and right boundaries of the $j$th cell for $j = 1, \ldots, N$ are $x_{j-1/2} = (j-1)\Delta x$ and $x_{j+1/2} = j\Delta x$ respectively. FV methods also use a scalar value $u_j(t)$ to represent the solution average within each cell where $u_j(t) := \int_{x_{j-1/2}}^{x_{j+1/2}} u(x, t) \, dx$. The standard FV equations for the time-derivative of $u_j$ in 1D and $u_{i,j}$ in 2D are simply discrete versions of the continuity equation:

$$\frac{\partial u_j}{\partial t} + \frac{f_{j+\frac{1}{2}} - f_{j-\frac{1}{2}}}{\Delta x} = 0 \quad \text{(3a)} \qquad \frac{\partial u_{i,j}}{\partial t} + \frac{f^x_{i+\frac{1}{2},j} - f^x_{i-\frac{1}{2},j}}{\Delta x} + \frac{f^y_{i,j+\frac{1}{2}} - f^y_{i,j-\frac{1}{2}}}{\Delta y} = 0. \tag{3b}$$

$f_{j+1/2}$ is the flux at the cell boundary $x_{j+1/2}$ and $f^x_{i+1/2,j}$ and $f^y_{i,j+1/2}$ are the aver-

age x-directed and y-directed fluxes through the right and top cell boundaries, e.g., $f^x_{i+1/2,j} := \frac{1}{\Delta y} \int_{y=y_{j-1/2}}^{y=y_{j+1/2}} \hat{\boldsymbol{x}} \cdot \boldsymbol{f}(x_{i+1/2}, y)\, dy$. In 1D, eq. (3a) can be derived by applying the integral $\int_{x_{j-1/2}}^{x_{j+1/2}} (...)\, dx$ to eq. (1) for all $j \in 1, \ldots, N$; a similar calculation in 2D gives eq. (3b). So long as $f_{j+1/2}$, $f^x_{i+1/2,j}$ and $f^y_{i,j+1/2}$ are exact for all $t$, then $u_j$ and $u_{ij}$ will be exact for all $t$. Thus, the key challenge for a FV scheme is to accurately reconstruct the flux at cell boundaries.

For the rest of this paper, we consider solvers that use MOL and the FV method. We will also assume that the ODE integration is stable; this can usually be done by using a strong stability preserving Runge Kutta (SSPRK) ODE integration method [12, 11] and choosing the timestep to satisfy a CFL condition. We also restrict ourselves to rectangular, periodic grids with uniform cell size.

**Conservation properties:** FV schemes conserve a discrete analogue $\sum_{j=1}^{N} u_j \Delta x$ of the continuous invariant $\int u\, dx$ by construction. In 1D, we can see this with a short proof: $d/dt \sum_{j=1}^{N} u_j \Delta x = \Delta x \sum_{j=1}^{N} \partial u_j / \partial t = -\sum_{j=1}^{N} (f_{j+1/2} - f_{j-1/2}) = f_{N+1/2} - f_{1/2}$. The rate of change of the discrete mass is equal to the flux of $u$ through the boundaries; in a periodic system this equals 0.

**Stability properties:** Although FV schemes inherit a discrete analogue of conservation of mass by construction, they do not automatically inherit discrete analogues of any of the stability properties of the continuous system eq. (1). Instead, FV methods ensure stability through careful choice of flux. The only known way of inheriting discrete analogues of all three stability properties of eq. (1) (non-increasing $\ell_p$-norm, non-increasing $\ell_\infty$-norm, and TVD) is to use a consistent monotone flux function (see [30] for definitions of consistency and monotonicity). An example of a monotone flux function for the linear advection equation $f = cu$ is the upwind flux; for non-linear $f(u)$ examples of monotone flux functions include the Godunov flux and the Lax-Friedrichs flux. Unfortunately, Godunov's famous theorem from 1959 implies that monotone schemes can be at most first-order accurate [10]; this means that while monotone schemes are great at stability, they are usually not very accurate. Fortunately, for a solver of eq. (1) to be stable it only has to inherit a discrete analogue of *one* of the three stability properties of the continuous equation [8]. This was one of the insights leading Van Leer's seminal paper introducing the MUSCL scheme [43]. MUSCL inherits a discrete analogue of the TVD property (which guarantees stability and prevents spurious oscillations by adding numerical diffusion to extremum and steep gradients) while retaining higher-order accuracy. Spurious oscillations are unphysical oscillations which develop around steep gradients [19] while numerical diffusion is implicit or explicit diffusion added to a high-order method, usually to preserve a stability property [28].

## 3.1 The Energy Method for Stability Analysis

As we learned in section 3, for a numerical method to be stable, it must inherit one or more of the stability properties of eq. (1). The energy method is a technique that analyzes whether a numerical method inherits a discrete analogue of the non-increasing $\ell_p$-norm property. $p = 2$ is usually chosen. Advantages of the energy method are that it can be used to analyze the stability of discrete methods for solving eq. (1) even when $\boldsymbol{f}(u)$ is non-linear, when BCs are non-periodic [8], and with certain systems of hyperbolic PDEs [23, 18]. Using the energy method, in the time-continuous limit a 1D discrete numerical algorithm for eq. (1) will be $\ell_2$-norm stable if $\frac{d}{dt} \sum_{j=1}^{N} (u_j)^2 \Delta x \leq 0$ for all $t$. Some simple algebra gives $\frac{d}{dt} \Delta x \sum_{j=1}^{N} u_j^2 / 2 = \Delta x \sum_{j=1}^{N} u_j \frac{\partial u_j}{\partial t}$. Using eq. (3a), this equals $-\sum_{j=1}^{N} u_j (f_{j+1/2} - f_{j-1/2})$. Performing summation by parts gives

$$\frac{d}{dt} \frac{\Delta x}{2} \sum_{j=1}^{N} (u_j)^2 = \sum_{j=1}^{N} f_{j+1/2}(u_{j+1} - u_j) \leq 0. \tag{4}$$

A discrete FV solver in 1D will be $\ell_2$-norm stable if eq. (4) is satisfied for all $t$. For non-periodic BCs eq. (4) includes a term which depends on the flux through the boundaries (see appendix B).

## 4 Global Stabilization of Flux Predicting FV Schemes

We now introduce 'global stabilization,' a technique that guarantees the $\ell_2$-stability of any FV scheme given by eq. (3a) or (3b). In section 6, we will discuss how to use global stabilization as a constraint on the model architecture of machine learned solvers. To derive this method in 1D with periodic

BCs, we begin with the energy method-based $\ell_2$-norm stability condition eq. (4). Let us now define $d\ell_2^{\text{old}}/dt := \sum_{j=1}^{N} f_{j+\frac{1}{2}}(u_{j+1} - u_j)$ as the original rate of change of the discrete $\ell_2$-norm, and $d\ell_2^{\text{new}}/dt$ as the desired rate of change of the discrete $\ell_2$-norm. We also define $\boldsymbol{u}_j := \{u_j\}_{j=1}^{N}$ as a vector representation of the discrete solution. We can change the time-derivative of the discrete $\ell_2$-norm from $d\ell_2^{\text{old}}/dt$ to $d\ell_2^{\text{new}}/dt$ by making the following transformation to $f_{j+1/2}$:

$$f_{j+\frac{1}{2}} \Rightarrow f_{j+\frac{1}{2}} + \frac{(d\ell_2^{\text{new}}/dt - d\ell_2^{\text{old}}/dt)G_{j+1/2}(\boldsymbol{u}_j)}{\sum_{k=1}^{N} G_{k+1/2}(\boldsymbol{u}_k)(u_{k+1} - u_k)} \tag{5}$$

for any scalar $d\ell_2^{\text{new}}/dt$ and any non-constant, finite function $G_{j+1/2}(\boldsymbol{u}_j)$ in which $\sum_{k=1}^{N} G_{k+1/2}(\boldsymbol{u}_k)(u_{k+1} - u_k) \neq 0$. As the reader can verify by plugging eq. (5) into eq. (4), eq. (5) modifies $f_{j+1/2}$ in a way that adds a constant $(d\ell_2^{\text{new}}/dt - d\ell_2^{\text{old}}/dt)$ to eq. (4) via cancellation of the denominator. Note that $G_{j+1/2}(\boldsymbol{u}_j)$ is a hyper parameter that determines how each $f_{j+1/2}$ is modified and $d\ell_2^{\text{new}}/dt$ is a user-defined quantity which sets the rate of change of the discrete $\ell_2$-norm. We want $d\ell_2^{\text{new}}/dt \leq 0$ for stability. A similar calculation in 2D reveals that the rate of change of the discrete $\ell_2$-norm is given by

$$\frac{d}{dt} \sum_{i,j} \frac{u_{i,j}^2}{2} \Delta x \Delta y = \Delta y \sum_{i,j} f_{i+\frac{1}{2},j}^x(u_{i+1,j} - u_{i,j}) + \Delta x \sum_{i,j} f_{i,j+\frac{1}{2}}^y(u_{i,j+1} - u_{i,j}) \leq 0. \tag{6}$$

We define $d\ell_2^{\text{old},x}/dt := \Delta y \sum_{i,j} f_{i+\frac{1}{2},j}^x(u_{i+1,j} - u_{i,j})$ and $d\ell_2^{\text{old},y}/dt := \Delta x \sum_{i,j} f_{i,j+\frac{1}{2}}^y(u_{i,j+1} - u_{i,j})$. Equation (6) will be satisfied if the following transformations are made to $f_{i+\frac{1}{2},j}^x$ and $f_{i,j+\frac{1}{2}}^y$:

$$f_{i+\frac{1}{2},j}^x \Rightarrow f_{i+\frac{1}{2},j}^x + \frac{(d\ell_2^{\text{new},x}/dt - d\ell_2^{\text{old},x}/dt)G_{i+1/2,j}^x(\boldsymbol{u}_{ij})}{\Delta y \sum_{k,l} G_{k+1/2,l}^x(\boldsymbol{u}_{kl})(u_{k+1,l} - u_{k,l})} \tag{7a}$$

$$f_{i,j+\frac{1}{2}}^y \Rightarrow f_{i,j+\frac{1}{2}}^y + \frac{(d\ell_2^{\text{new},y}/dt - d\ell_2^{\text{old},y}/dt)G_{i,j+1/2}^y(\boldsymbol{u}_{ij})}{\Delta x \sum_{k,l} G_{k,l+1/2}^y(\boldsymbol{u}_{kl})(u_{k,l+1} - u_{k,l})} \tag{7b}$$

for any scalars $d\ell_2^{\text{new},x}/dt$ and $d\ell_2^{\text{new},y}/dt$ where $d\ell_2^{\text{new},x}/dt + d\ell_2^{\text{new},y}/dt \leq 0$ and any non-constant, finite functions $G_{i+1/2,j}^x(\boldsymbol{u}_{ij})$ and $G_{i,j+1/2}^y(\boldsymbol{u}_{ij})$ for which $\sum_{k,l} G_{k+1/2,l}^x(\boldsymbol{u}_{kl})(u_{k+1,l} - u_{k,l}) \neq 0$ and $\sum_{k,l} G_{k,l+1/2}^y(\boldsymbol{u}_{kl})(u_{k,l+1} - u_{k,l}) \neq 0$. Equations (5), (7a) and (7b) ensure for scalar conservation form PDEs in 1D and 2D that the discrete $\ell_2$-norm will be non-increasing in the time-continuous limit.

In our experiments we set $G_{j+1/2}(\boldsymbol{u}_j) = (u_{j+1} - u_j)$, $G_{i+1/2,j}^x(\boldsymbol{u}_{ij}) = (u_{i+1,j} - u_{i,j})$, and $G_{i,j+1/2}^y(\boldsymbol{u}_{ij}) = (u_{i,j+1} - u_{i,j})$. These choices have a simple physical interpretation: they correspond to the addition of a spatially constant diffusion coefficient everywhere in space [27]. Possible alternatives include setting $G_{j+1/2}(\boldsymbol{u}_j) = (u_{j+1} - u_j)^\beta$ for $\beta > 1$ or $G_{j+1/2}(\boldsymbol{u}_j) = \alpha_{j+1/2}(u_{j+1} - u_j)$ for $\alpha_{j+1/2} \in \mathbb{R}$. Choosing large $\beta$ increases the amount of numerical diffusion added at discontinuities and decreases the amount of diffusion added in smooth regions, while $\alpha_{j+1/2}$ is a spatially dependent scalar which determines a spatially varying distribution of added numerical diffusion.

Global stabilization allows the user to control the rate of change of the $\ell_2$-norm; this can either stabilize an unstable method or reduce or eliminate numerical diffusion from a stable method.

Figure 1 demonstrates how global stabilization can stabilize an unstable scheme. On the inviscid Burgers equation the centered flux $f_{j+1/2} = (u_j^2 + u_{j+1}^2)/4$, shown in red, is unstable and inaccurate. We apply global stabilization to the centered flux with $d\ell_2^{\text{new}}/dt = 0$, shown in blue. This leads to exact conservation of the $\ell_2$-norm and a stable numerical method. Our initial condition is $u(x) = \sin x$.

Note that the globally stabilized centered flux solution in fig. 1 conserves both the discrete mass and the discrete $\ell_2$-norm, but does not maintain a discrete analogue of the total variation diminishing (TVD) property [8] of the scalar Burgers equation. As a result, the globally stable solution permits high-$k$ oscillations to develop; these spurious oscillations are often seen in schemes that do not have enough numerical diffusion to damp high-$k$ modes that develop near steep gradients [36].

Figure 2 demonstrates how global stabilization can reduce or eliminate numerical damping from a stable scheme. We solve the 2D incompressible Euler equations in vorticity form, given by

$$\frac{\partial \chi}{\partial t} + \boldsymbol{\nabla} \cdot (\boldsymbol{u}\chi) = 0, \qquad \boldsymbol{u} = \boldsymbol{\nabla}\psi \times \hat{e}_z, \qquad -\boldsymbol{\nabla}^2\psi = \chi. \tag{8}$$

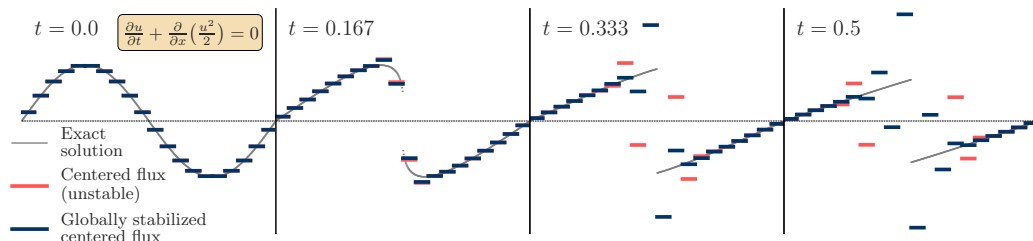

Figure 1: Global stabilization turns an unstable solver into a stable solver. While centered flux $f_{j+1/2} = (u_j^2 + u_{j+1}^2)/4$ is an unstable choice of flux (red) on the inviscid Burgers equation and blows up by $t = 0.5$, global stabilization (blue) ensures that the discrete $\ell_2$-norm is conserved.

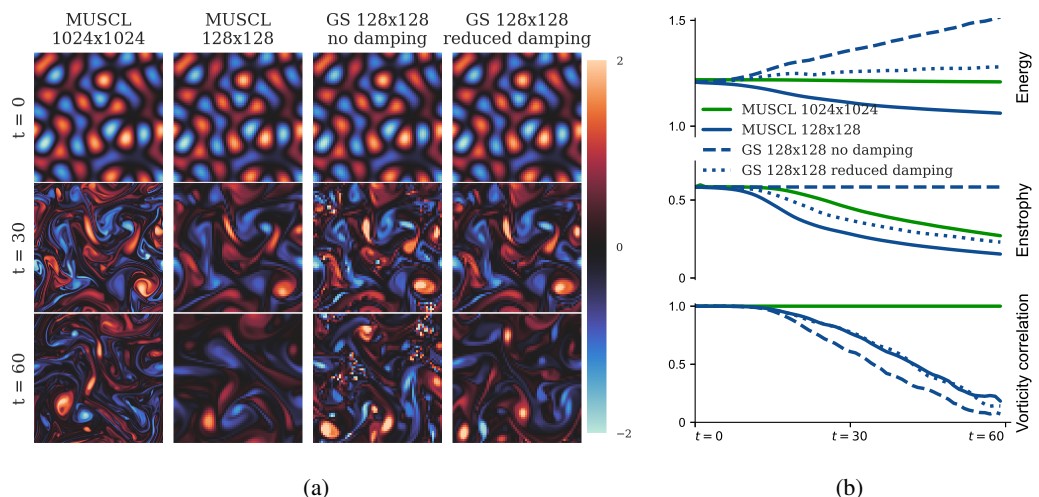

(a)                                                                      (b)

Figure 2: Applying global stabilization to a stable FV scheme can reduce or eliminate numerical diffusion, but at the cost of introducing spurious high-$k$ oscillations. (a) Images of the vorticity $\chi$ evolving under the incompressible Euler equations. The first and second columns show the baseline MUSCL scheme at high and low resolution. The third and fourth columns show the baseline MUSCL scheme with global stabilization (GS), either with no numerical damping or with numerical damping reduced by 75%. (b) Energy, enstrophy, and vorticity correlation over time. We use vorticity correlation as a benchmark measure of accuracy.

Our baseline choice of flux is the second-order TVD MUSCL scheme with monotonized central (MC) flux limiters [40, 43]. We use a linear finite element (FE) solver for the poisson equation [1] and a strong stability preserving RK3 ODE integrator [12]. Note that eq. (8) exactly conserves both the energy $\frac{1}{2} \int \boldsymbol{u}^2 \, dx \, dy$ and the enstrophy $\int \chi^2 \, dx \, dy$ [37].

In each column of fig. 2a, we see the time-evolution of the vorticity $\chi$ according to four schemes. The baseline MUSCL schemes (1st and 2nd columns) decay the discrete $\ell_2$-norm, while the MUSCL schemes with global stabilization (GS, 3rd and 4th colums) either exactly conserve $\ell_2$-norm by setting $d\ell_2^{\text{new},x}/dt = d\ell_2^{\text{new},y}/dt = 0$ (no damping) or reduce the rate of numerical diffusion by 75% by setting $d\ell_2^{\text{new},x}/dt = \frac{1}{4} d\ell_2^{\text{old},x}/dt$ and $d\ell_2^{\text{new},y}/dt = \frac{1}{4} d\ell_2^{\text{old},y}/dt$ (reduced damping). In the 3rd column, we again find that ensuring $\ell_2$-norm conservation introduces spurious high-$k$ oscillations. In fig. 2b, bottom row, we plot the vorticity correlation between the high resolution baseline and each of the four schemes. Vorticity correlation has been used previously as a benchmark measure of accuracy for eq. (8) [22]. We find that global stabilization with no damping underperforms relative to the baseline at the same resolution, while global stabilization with reduced damping performs similarly. In fig. 2b, middle and top rows, we plot the discrete enstrophy $\sum_{i,j} \int \chi_{i,j}^2 \Delta x \Delta y$ and discrete energy $\frac{1}{2} \sum_{i,j} \int (\boldsymbol{u}_{ij})^2 \Delta x \Delta y$. The baselines decay energy and enstrophy, while the globally stabilized schemes do not conserve energy and either exactly conserve enstrophy (no damping) or decay enstrophy (reduced damping).

# 5 Global Stabilization of MOL Schemes with Arbitrary Time-Derivative

In section 4, we considered schemes that predict the flux $f$ at cell boundaries. Using the energy method, we found that we could adjust the flux prediction to ensure global stability. However, some machine learned PDE solvers may use an alternative form for the time-derivative which does not involve predicting the flux at cell boundaries. Thus, we now consider the more general problem of how to stabilize MOL-based solvers for eq. (1) with arbitrary time-derivative. Suppose that the rate of change of the cell average $u_j$ in 1D or $u_{i,j}$ in 2D is given by

$$\frac{\partial u_j}{\partial t} = N_j(\boldsymbol{u}_j) \qquad (9\text{a}) \qquad\qquad \frac{\partial u_{i,j}}{\partial t} = N_{i,j}(\boldsymbol{u}_{ij}) \qquad (9\text{b})$$

where $N_j(\boldsymbol{u}_j)$ and $N_{i,j}(\boldsymbol{u}_{ij})$ are arbitrary functions and $\boldsymbol{u}_j$ and $\boldsymbol{u}_{ij}$ are again vector representations of the discrete solution. Note that eqs. (9a) and (9b) do not guarantee mass conservation by construction. Ensuring stability and mass conservation therefore requires modifying $N_j$ and $N_{i,j}$. In 1D, we have $\Delta x \sum_{j=1}^{N} \frac{\partial u_j}{\partial t} = \Delta x \sum_j N_j = 0$ and $\Delta x \sum_{j=1}^{N} u_j \frac{\partial u_j}{\partial t} = \Delta x \sum_j u_j N_j \le 0$. These imply that the discrete mass will be conserved if $\langle \boldsymbol{N}_j \rangle := \sum_{j=1}^{N} N_j = 0$ and the discrete $\ell_2$-norm will decay if $\langle \boldsymbol{u}_j | \boldsymbol{N}_j \rangle := \sum_{j=1}^{N} u_j N_j \le 0$. The bracket notation $\langle \dots \rangle$ denotes the mean value over the domain while the inner product notation $\langle \dots | \dots \rangle$ denotes a sum over all domain cells. These conditions will be satisfied if the following transformation is applied to $N_j$:

$$\boldsymbol{U}_j := \boldsymbol{u}_j - \langle \boldsymbol{u}_j \rangle \quad\quad \boldsymbol{M}_j := \boldsymbol{N}_j - \langle \boldsymbol{N}_j \rangle \quad\quad \boldsymbol{N}_j \Rightarrow \boldsymbol{M}_j + \frac{\frac{d\ell_2^{\text{new}}}{dt} \boldsymbol{G}_j(\boldsymbol{u}_j)}{\langle \boldsymbol{U}_j | \boldsymbol{G}_j(\boldsymbol{u}_j) \rangle} - \frac{\langle \boldsymbol{U}_j | \boldsymbol{M}_j \rangle}{\langle \boldsymbol{U}_j | \boldsymbol{U}_j \rangle} \boldsymbol{U}_j \quad (10)$$

for any $d\ell_2^{\text{new}}/dt \le 0$ and any smooth function $\boldsymbol{G}_j(\boldsymbol{u}_j)$ where $\langle \boldsymbol{G}_j(\boldsymbol{u}_j) \rangle = 0$ and $\langle \boldsymbol{G}_j(\boldsymbol{u}_j) | \boldsymbol{U}_j \rangle \ne 0$. The choice $G_j(\boldsymbol{u}_j) = (\nabla^2 u)_j = u_{j+1} + u_{j-1} - 2u_j$ adds a spatially constant diffusion coefficient.

# 6 Stable Machine Learned PDE Solvers

The purpose of this paper is to demonstrate how to design stable *machine learned solvers*. Global stabilization can be applied to (a) 'hybrid' MOL-based machine learned solvers for eq. (1) (b) that use ML to approximate the divergence term $\nabla \cdot \boldsymbol{f}(u)$ in the time-continuous limit.

Regarding (a), the defining feature of a hybrid machine learned solver is that it inherits one or more of the properties of classical numerical methods. See section 7 for examples of papers that use hybrid solvers. Usually this involves MOL, i.e., discretizing the domain into a number of grid cells and using some sort of time-stepping procedure or ODE integration to advance the solution in time.

Regarding (b), approximating the divergence term is usually the most difficult element of a numerical method, so it is fairly common to replace this term with a machine-learned approximation. Some hybrid solvers may use the FV method and use ML to approximate the flux across cell boundaries $f_{j+1/2}$. Other hybrid solvers may use the more general time-derivative function eq. (9a). Note that global stabilization can also be used when the RHS of eq. (1) is non-zero (see appendix A).

Recall that global stabilization requires setting the value of $d\ell_2^{\text{new}}/dt$. According to eq. (4), for stability we want the discrete $\ell_2$-norm of the exact solution to be non-increasing for all $t$. Thus, in algorithm 1 we propose a practical method for choosing $d\ell_2^{\text{new}}/dt$ when applying global stabilization to machine learned PDE solvers that satisfy the conditions (a) and (b). Algorithm 1 can be used to ensure stability of machine learned solvers that predict $f_{j+1/2}$ in eq. (3a) or to ensure mass conservation and stability of solvers that use equation eq. (9a). Algorithm 1 does not change the output of the machine learned PDE solver if the solver tries to decay the discrete $\ell_2$-norm, but sets $d\ell_2^{\text{new}}/dt = 0$ if the solver tries to increase the discrete $\ell_2$-norm. Intuitively, algorithm 1 is an error correcting algorithm that adjusts the output of the machine learned solver only if that output moves the solution towards instability.

## 6.1 Towards a Deeper Understanding of Global Stabilization

Readers familiar with classical numerical methods, which ensure stability via locally-derived constraints on the flux $f_{j+1/2}$, might ask: why put global, rather than local, constraints on the flux $f_{j+1/2}$?

It is of course *possible* to guarantee stability of machine learned numerical methods by putting local constraints on the flux. One could, for example, develop a TVD method by applying a flux limiter

---

**Algorithm 1** A stable machine learned MOL-based PDE solver in 1D

---

1: **Inputs**: Initial condition $\{u_j(t_0)\}_{j=1}^{N_x}$, ODE integrator, ML predictor for $f_{j+\frac{1}{2}}$ or $N_j$

2: **while** $t < T_f$ **do**

3:     Choose $\Delta t$, compute $\{f_{j+\frac{1}{2}}\}_{j=1}^{N_x}$ or $\{N_j\}_{j=1}^{N_x}$ using ML predictor

4:     **if** using eq. (3a) **then**

5:       **if** $d\ell_2^{\text{old}}/dt = \sum_j f_{j+\frac{1}{2}}(u_{j+1} - u_j) > 0$ **then**

6:         Set $\{f_{j+\frac{1}{2}}\}_{j=1}^{N_x}$ according to eq. (5) with $d\ell_2^{\text{new}}/dt = 0$

7:     **else if** using eq. (9a) **then**

8:       **if** $\langle M_j | u_j \rangle \leq 0$ **then**

9:         Set $N_j = M_j$

10:      **else**

11:        Set $N_j$ according to eq. (10) with $d\ell_2^{\text{new}}/dt = 0$

12:    Advance time $t$ by $\Delta t$ and state $\{u_{j+1}\}_{j=1}^{N_x}$ according to ODE integrator

13: **Output:** $\{u_j(T_f)\}_{j=1}^{N_x}$

---

to a machine learned solver that predicts $f_{j+1/2}$. The problem with doing this is that (a) the goal of machine learned solvers is to use fewer computational resources than classical numerical methods, which requires solving the equations at coarser resolution, i.e., larger $\Delta x$ (see the discussion of LES models in section 7), (b) at coarse resolution a high proportion of grid cells are either extremum or have sharp gradients, (c) local constraints like flux limiters add numerical diffusion to extremum and sharp gradients, and (d) the magnitude of numerical diffusion goes like $(\Delta x)^2$ [27]. The implication of (a), (b), (c), and (d) is that TVD-stable machine learned numerical methods operating at coarser resolution than classical solvers will add large amounts of numerical diffusion to many of the grid cells which will rapidly degrade the accuracy of the solution. Improving accuracy at coarse resolution requires a solver that has the freedom to make flexible predictions; simultaneously ensuring stability requires finding a way to do so while adding less numerical diffusion than standard techniques. Because algorithm 1 uses global constraints, it is able maintain flexibility while adding the *minimum* amount of numerical diffusion necessary to ensure $\ell_2$-norm stability.

In fact, the whole point of the global stabilization method is that it can guarantee the stability of a solver *without* degrading the accuracy of an already-accurate solver. This is possible because (a) numerical diffusion is only added if the machine learned solver violates the non-increasing $\ell_2$-norm property of the solution, (b) a highly accurate machine learned solver is unlikely to violate this property within its training distribution, and (c) even if it does so the additional numerical diffusion is the minimum required to correct the violation. (a) and (c) are implied by algorithm 1, while (b) is discussed in appendix C. In other words, for a well-engineered machine learned PDE solver we can expect the effects of global stabilization to be infrequent, small, and applied only when necessary. This is what we find in fig. 3 when we apply global stabilization to a machine learned PDE solver trained to find an accurate solution to the 1D advection equation $f(u) = u$ by predicting $f_{j+1/2}$ at each cell boundary. Figure 3 shows that while global stabilization (ML GS) has a negligible impact on the accuracy of the machine learned solver (ML), using a TVD flux limiter to guarantee stability (ML MC Limiter) leads to much worse accuracy. Further details are in appendix D.

## 7 Related Work

**LES models and backscattering**: The objective of large eddy simulation (LES) is identical to that of many machine learned numerical solvers: both attempt to find an accurate approximation to the solution of the PDE with fewer computational resources than classical numerical methods. Both also attempt to do so without resolving the smallest scales of the problem, relying on either an explicit or implicit subgrid model to do so [2, 45, 22, 39, 42, 25, 34, 14, 15, 29, 3, 44, 32]. Of particular relevance to the stability of subgrid models (both in LES and ML) are the concepts of 'forward-scatter' and 'backscatter'. In 2D LES turbulence, forward-scattering involves the transfer of enstrophy from resolved to unresolved scales, while backscattering involves the transfer of enstrophy from unresolved to resolved scales. Analysis across a wide range of flows demonstrates two important facts [35]. First, to be accurate a subgrid model must allow both forward-scatter and backscatter. In

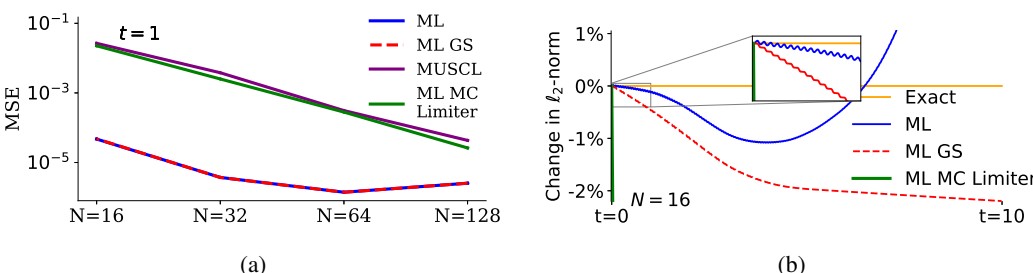

Figure 3: (a) Mean squared error (MSE) for $t = 1$ as a function of $N$ for four schemes used to solve 1D advection. (b) The percent change in the discrete $\ell_2$-norm for a single example drawn from the training distribution.

the context of scalar hyperbolic PDEs, this means that to be accurate a subgrid model must allow a discrete analogue of the entropy inequality in eq. (2) to be locally violated. Second, averaged over the entire domain there is always more forward-scatter than backscatter. If on average there were more backscatter than forward-scatter, then the subgrid model would be unstable [14]. Global stabilization can thus be interpreted as a way of constraining a subgrid model to ensure that on average there is always at least as much forward-scatter as backscatter.

**Machine learned finite volume solvers**: [2, 45, 22] use machine learned finite volume solvers to solve a variety of 1D and 2D PDEs. Almost all of these PDEs can be written in conservation form with added diffusion and forcing terms. These 'hybrid' solvers conserve mass by construction but not the discrete $\ell_2$-norm and therefore do not guarantee stability; instead, they promote stability by unrolling the loss function over multiple timesteps. [22] trains a hybrid solver for the 2D incompressible Euler equations that "remains stable during long simulations." This impressive result is likely facilitated by the addition of physical diffusion to the PDE, which decays the $\ell_2$-norm at each timestep.

**Other machine learned solvers**: [25, 42, 34, 22] use convolutional neural networks to correct errors in low-resolution simulations; these hybrid solvers promote stability and improve accuracy by unrolling the loss function over multiple timesteps. [39] solves 2D and 3D hyperbolic PDEs using the 'fully learned' update equation $u_{i,j}(t + \Delta t) = u_{i,j}(t) + N_{i,j}(u_{i,j}(t), \Delta t))$ where $N_{i,j}$ is a the output of a convolutional neural network; this update equation is similar to eq. (9b) except with a discrete-time update instead of continuous-time ODE integration. [39] attempts to ensure stability by adding noise to the training distribution and by using very large timesteps. For scalar conservation form PDEs, this fully learned update equation will be stable if $\langle \boldsymbol{N} \rangle = 0$ and $\langle \boldsymbol{u} | \boldsymbol{N} \rangle + \langle \boldsymbol{N} | \boldsymbol{N} \rangle \leq 0$. [4] argues that instability in machine learned iterative numerical algorithms arises due to a *distribution shift* where the distribution of training data differs from the outputs of the solver during inference due to small errors that accumulate over time. [4] uses the update equation $u_{i,j}(t + \ell\Delta t) = u_{i,j}(t) + \ell\Delta t N_{i,j}^{\ell}$ for $1 \leq \ell \leq K$ where $N_{i,j}^{\ell}$ is the output of a message passing graph neural network that predicts the next $K$ timesteps. [4] attempts to ensure stability by modifying the loss function, adding random noise, and by predicting multiple timesteps into the future. A variety of papers have attempted to promote stability of dynamical systems that result from data-driven reduced order models, including by adding sparsity-promoting priors to a loss function [20, 9] and by constraining the eigenvalues of a learned Koopman operator [33].

**KEP schemes**: Kinetic energy preserving (KEP) and entropy preserving (EP) schemes can be used in the numerical study of hyperbolic equations. Like global stabilization, KEP and EP schemes rely on the energy method for stability analysis and use summation by parts [17, 18, 6]. Unlike global stabilization, these schemes construct locally conservative algorithms which add just enough numerical damping at shocks to eliminate spurious oscillations.

## 8  Limitations

There are four main limitations of our work. First, we only consider rectangular grids, periodic BCs, and *scalar* hyperbolic PDEs in conservation form. In particular, we do not consider *systems* of hyperbolic PDEs. Although many physically relevant equations can be written as scalar hyperbolic PDEs – including Hamiltonian systems, the incompressible Euler equations, and the Vlasov-Poisson equation

303  – many more are systems of hyperbolic PDEs – including the compressible Euler equations, the
304  magnetohydrodynamic (MHD) equations, the Einstein field equations, the shallow-water equations,
305  the Navier-Stokes equations, and the Vlasov-Maxwell equations. Fortunately, the energy method can
306  be extended to non-periodic BCs (see appendix B) and certain systems of PDEs [23, 18]. Furthermore,
307  it is standard practice in the numerical methods community to first use the scalar conservation law
308  eq. (1) to introduce a new method before later extending the method to systems of PDEs [7]. We
309  anticipate that our method could be extended to many physically relevant systems of hyperbolic PDEs
310  in a manner similar to KEP and EP schemes [17, 18].

311  Second, our method works with MOL in the continuous-time limit. The timestep $\Delta t$ must be chosen
312  to satisfy a CFL condition and be small enough to ensure accuracy of the ODE integration. Some
313  machine learned solvers use large $\Delta t$ or predict multiple timesteps at once or don't use MOL;
314  algorithm 1 cannot be used to stabilize these solvers.

315  Third, while global stabilization is designed to solve the problem of ensuring stability of machine
316  learned solvers for eq. (1) without degrading accuracy, it does not solve the problem of *finding*
317  accurate machine learned solvers for eq. (1). Algorithm 1 prevents instability by adjusting the
318  time-derivative if the solver makes an $\ell_2$-norm increasing violation, but a solver which frequently
319  commits such violations is likely to perform poorly. Alternatively, a solver could make no $\ell_2$-norm
320  increasing violations but decay the $\ell_2$-norm too quickly. Or, it could decay the $\ell_2$-norm at the correct
321  rate but give inaccurate results. Building accurate, fast, and robust machine learned PDE solvers will
322  require not only well-designed numerical methods but also well-engineered learning systems which
323  consistently make accurate predictions about the time evolution of the solution.

324  Fourth, for some scalar hyperbolic PDEs a solver might be stable according to the definition in
325  section 2 but not result in a physically meaningful solution as $t \to \infty$ unless additional physical
326  constraints are satisfied. In appendix E, we give an example of this issue and illustrate how this forth
327  limitation might be addressed by demonstrating that for some equations it may be possible to develop
328  global stabilization schemes that enforce additional conservation laws.

# 9  Conclusion

330  Stability is a very desirable property of a PDE solver. Machine learned PDE solvers have tried a
331  variety of techniques to encourage stability (see section 7). To some extent, these techniques have
332  been successful, as high-performing solvers have demonstrated the ability to give stable and accurate
333  predictions for hundreds or thousands of timesteps. However, none of these techniques are capable of
334  *guaranteeing* stability.

335  In this paper, we show how to design machine learned PDE solvers for scalar hyperbolic PDEs
336  that are stable by construction. The main result of our paper is the 'global stabilization' technique.
337  This can be used as an error-correcting algorithm to guarantee both mass conservation and $\ell_2$-norm
338  stability of hybrid machine learned PDE solvers (see section 6), even when the time-derivative is an
339  arbitrary function (see section 5).

340  As we have seen, it is impossible to design highly accurate numerical methods that inherit all of
341  the properties of eq. (1); this is implied by Godunov's theorem (see section 3). We have also seen
342  that it is often not even *desirable* for a numerical method to inherit the properties of the continuous
343  equation, as doing so can significantly degrade the quality of the solution (see the discussion of
344  backscattering in section 7 as well as fig. 2). The conclusion is that designers of numerical methods
345  must determine which properties of the continuous system should be preserved by the discrete system
346  and which properties either cannot be preserved or degrade the accuracy of the discrete system. Global
347  stabilization preserves conservation of mass and $\ell_2$-norm stability, but allows the time-derivative
348  to depend on the global solution which violates the property of hyperbolic PDEs that information
349  propagates at finite speed. While we would prefer our numerical methods to maintain this property if
350  possible, the benefit of not doing so is that global stabilization can ensure stability without degrading
351  the accuracy of an already-accurate machine learned solver (see section 6.1 and appendix D).

352  We believe that for machine learned PDE solvers to have real-world impact, they must be sufficiently
353  robust and reliable to be trusted. Global stabilization, by guaranteeing stability, is a step towards the
354  development of robust and reliable machine learned PDE solvers which could have real-world impact.

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

# A    Conservation Form PDEs with Nonzero Right Hand Side

Classical methods for ensuring stability work even when the right hand side (RHS) of eq. (1) is non-zero. Thus, it should not be surprising that the global stabilization method can be used even

when the right hand side (RHS) of eq. (1) is non-zero. This is because the only term which usually contributes to numerical instability is the divergence term. Using a stable method to approximate eq. (1) is sufficient to ensure stability so long as the RHS terms don't contribute to numerical instability. Usually they do not.

For example, suppose we have the model equation $\partial u/\partial t + \boldsymbol{\nabla} \cdot \boldsymbol{f}(u) = D\boldsymbol{\nabla}^2 u + F(\boldsymbol{x}, t)$ where $D \in \mathbb{R}$ is a non-negative diffusion coefficient and $F(\boldsymbol{x}, t)$ is a forcing function. We can approximate the diffusion term using a standard approximation of the laplacian operator, we know that this can only decrease the $\ell_2$-norm. Likewise, we can approximate the forcing term using a quadrature; this forcing term can contribute to *physical* instability but will not contribute to *numerical* instability. The only term which contributes to numerical instability is the divergence term; we can apply global stabilization to the approximation of this term.

# B  Generalization to Non-Periodic BCs and Non-Uniform Grid Spacing

We begin by modifying eq. (5) for non-periodic boundary conditions. We begin by calculating the $\ell_2$-norm stability condition according to the energy method:

$$\frac{d}{dt}\Delta x \sum_{j=1}^{N} u_j^2/2 = \Delta x \sum_{j=1}^{N} u_j \frac{\partial u_j}{\partial t} = -\sum_{j=1}^{N} u_j(f_{j+1/2} - f_{j-1/2}).$$

Next, we perform summation by parts:

$$\frac{d}{dt}\frac{\Delta x}{2} \sum_{j=1}^{N} (u_j)^2 = \sum_{j=1}^{N-1} f_{j+1/2}\big(u_{j+1} - u_j\big) \leq f_{N+1/2}u_N - f_{1/2}u_1.$$

This stability condition will be satisfied if the following transformation is made to $f_{j+1/2}$:

$$f_{j+\frac{1}{2}} \Rightarrow f_{j+\frac{1}{2}} + \frac{(d\ell_2^{\text{new}}/dt - d\ell_2^{\text{old}}/dt)G_{j+1/2}(\boldsymbol{u}_j)}{\sum_{k=1}^{N-1} G_{k+1/2}(\boldsymbol{u}_k)(u_{k+1} - u_k)} \tag{11}$$

where $d\ell_2^{\text{new}}/dt \leq f_{N+1/2}u_N - f_{1/2}u_1$ and we define $d\ell_2^{\text{old}}/dt \coloneqq \sum_{j=1}^{N-1} f_{j+\frac{1}{2}}(u_{j+1} - u_j)$.

Next, we modify eq. (5) for non-uniform grid spacing. We begin by calculating the $\ell_2$-norm stability condition according to the energy method:

$$\frac{d}{dt}\frac{1}{2} \sum_{j=1}^{N} \Delta x_j (u_j)^2 = \sum_{j=1}^{N} \Delta x_j u_j \frac{\partial u_j}{\partial t} = -\sum_{j=1}^{N} u_j(f_{j+1/2} - f_{j-1/2}) = \sum_{j=1}^{N} f_{j+1/2}\big(u_{j+1} - u_j\big) \leq 0$$

As we can see, the $\ell_2$-norm stability condition is unchanged for non-uniform grid spacing. Thus, eq. (5) is unchanged for non-uniform grid spacing.

Similar calculations can be performed to generalize the 2D expressions eqs. (7a) and (7b) to non-perioidic boundary conditions and non-uniform grid spacing.

# C  Coarse Graining and the $\ell_2$-Norm of the Training Data

The $\ell_2$-norm of the continuous exact solution to eq. (1) $u^{\text{exact}}(x, t)$ has non-increasing $\ell_2$-norm $\int_0^L (u^{\text{exact}}(x, t))^2 dx$. It turns out that the coarse-grained exact solution $u_j^{\text{exact}}(t) = \int_{x_{j-1/2}}^{x_{j+1/2}} u^{\text{exact}}(x, t)dx$ almost always has a non-increasing discrete $\ell_2$-norm $\sum_{j=1}^{N}(u_j^{\text{exact}}(t))^2 \Delta x$ as well. For linear $f(u)$ (i.e., the advection equation) the discrete $\ell_2$-norm of the exact solution can be, depending on the initial conditions, either (a) constant (see, for example, fig. 3b) (b) oscillatory (see, for example, fig. 6 of [45]) or (c) monotonically decreasing with high probability (see, for example, fig. 7 of [45]). For non-linear $f(u)$, the continuous solution $u^{\text{exact}}(x, t)$ develops high-$k$ modes and/or structures on a scale smaller than the grid size. These modes cannot be represented by the scalar $u_j^{\text{exact}}(t)$ and are replaced via coarse-graining by a low-dimensional representation of the solution which has lower $\ell_2$-norm with high probability. The result of coarse graining is that for

non-linear $f(u)$ the discrete $\ell_2$-norm of the exact solution is (d) monotonically decreasing with high probability (see, for example, fig. 2b).

We assume that the training data used to train a machine learned solver is the coarse-grained exact solution. We can expect that the rate of change of the discrete $\ell_2$-norm of the machine learned solution will be equal to the rate-of-change of the discrete $\ell_2$-norm of the training data plus $\epsilon$, where $\epsilon$ is some small error.

For (c) and (d), the discrete $\ell_2$-norm of the training data is monotonically decreasing, so we can expect a machine learned solver to also have decreasing discrete $\ell_2$-norm with high probability so long as $\epsilon$ is small. For (a), the discrete $\ell_2$-norm of the training data is constant and so we can expect a machine learned solver to have non-increasing discrete $\ell_2$-norm when $\epsilon < 0$ and increasing discrete $\ell_2$-norm when $\epsilon > 0$. Although for (a) a machine learned solver may frequently increase the $\ell_2$-norm, this increase is likely to be small so long as $\epsilon$ is likely to be small (see, for example, fig. 3b). For (b), the discrete $\ell_2$-norm of the training data oscillates and so a machine learned solver is likely to increase the discrete $\ell_2$-norm.

In summary, for non-linear $f(u)$ a machine learned solver is unlikely to increase the discrete $\ell_2$-norm within its training distribution. For linear $f(u)$, so long as the discrete $\ell_2$-norm of the training data doesn't oscillate in time, we can expect a machine learned solver either to be unlikely to increase the $\ell_2$-norm or to do so by only a small amount $\epsilon$.

## D  Global Stabilization of Machine Learned Solver for 1D Advection

We apply global stabilization to a machine learned solver for the 1D advection equation $\frac{\partial u}{\partial t} + c\frac{\partial u}{\partial x} = 0$ for $c \in \mathbb{R}$. Our choice of solver uses a convolutional neural network (CNN) to predict coefficients of a stencil of width 4 which reconstructs the solution $u_{j+1/2}$ and flux $f_{j+1/2} = cu_{j+1/2}$ at each cell boundary at each timestep; this so-called 'data-driven discretization' approach was introduced in [2]. We use periodic boundary conditions on the domain $x \in [0, 1]$ with $N$ grid cells and uniform cell width $\Delta x = 1/N$ and set $c = 1$.

Both the training data and the test data are given by the coarse-grained exact solution $u_j(t) = \int_{x_{j-1/2}}^{x_{j+1/2}} u^{\text{exact}}(x, t)dx$ where $u^{\text{exact}}(x, t)$ is known analytically using $u^{\text{exact}}(x, t) = u_0(x - ct)$ and $u_0(x)$ is the initial condition at $t = 0$. The initial condition is drawn from a sum-of-sines distribution

$$u_0(x) = \sum_{i=1}^{N^{\text{modes}}} A_i \sin\left(2\pi k_i + \phi_i\right)$$

where $N^{\text{modes}} \sim \{2, 3, 4, 5, 6\}$ and $k_i \sim \{0, 1, 2, 3\}$ are uniform draws from a set while $A_i \sim [-0.5, 0.5]$ and $\phi_i \sim [0, 2\pi]$ are draws from uniform distributions. The loss function $L$ is given by computing the mean squared error (MSE) unrolled over $N^{\text{unroll}} = 8$ timesteps [42]:

$$L = \frac{\Delta x}{N^{\text{unroll}}} \sum_{k=1}^{N^{\text{unroll}}} \sum_{j=1}^{N} \left(u_j(t + k\Delta t) - u_j^{\text{exact}}(t + k\Delta t)\right)^2.$$

We use a SSPRK3 ODE integrator [12] and choose the timestep $\Delta t$ using a CFL condition with a safety factor of 0.1. Our training data uses 200 samples from $t \in [0, 1]$. We train with a batch size of 8 and use the ADAM optimizer [21] for 1000 training iterations with a learning rate of $3 \times 10^{-3}$ followed by 1000 training iterations with a learning rate of $3 \times 10^{-4}$. Our CNN has three convolutional layers of width 32, kernel size 5, and ReLU non-linearity followed by a linear convolutional output with kernel size 4 for each of the 4 stencil coefficients at each cell boundary. We also ensure that our stencil coefficients sum to 1 at each cell boundary.

Figure 3a shows the MSE for $0 < t < 1$ as a function of the number of grid cells $N$ for four schemes used to solve the 1D advection equation: the baseline MUSCL scheme with monotonized central (MC) flux limiters (MUSCL), the original machine learned solver (ML), the machine learned solver with global stabilization (ML GS), and the machine learned solver with a flux limiter (ML MC Limiter). Our test set is the average over 50 data points drawn from the same distribution as the training set. We see that the MSE of the globally stabilized solver is almost identical to the MSE of the original machine learned solver, while using a TVD flux limiter to stabilize the solver leads to a

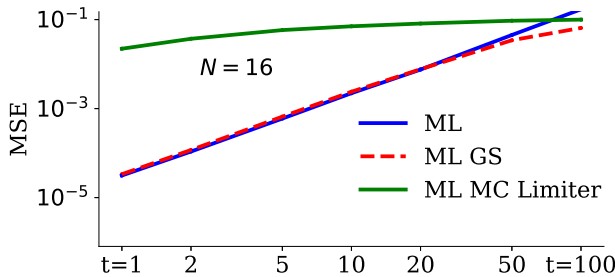

Figure 4: MSE for $N = 16$ as a function of time. Global stabilization has a negligible impact on the accuracy of the machine learned solver for small $t$ and improves accuracy at large $t$.

MSE which is significantly worse than the original machine learned solver and a MSE which is only slightly better than the baseline MUSCL scheme.

Figure 3b shows the percent change in the discrete $\ell_2$-norm for a single example drawn from the training distribution. While the exact solution $u_j^{\text{exact}}$ has constant discrete $\ell_2$-norm, the machine learned solver allows the discrete $\ell_2$-norm to both increase and decrease. The globally stabilized machine learned solver, however, can only decrease the $\ell_2$-norm. Meanwhile, the flux-limited machine learned solver rapidly decays the discrete $\ell_2$-norm.

Note that in fig. 3b the exact solution has a constant discrete $\ell_2$-norm. While algorithm 1 sets $d\ell_2^{\text{new}}/dt \leq 0$, for the 1D advection equation with a sum-of-sines initial condition we are able to set $d\ell_2^{\text{new}}/dt = 0$ at each timestep because the exact solution has constant discrete $\ell_2$-norm. Instead, to illustrate the properties of algorithm 1 we set $d\ell_2^{\text{new}}/dt \leq 0$.

Figure 4 shows the MSE for $N = 16$ as a function of time $t$ for three of the schemes used to solve the 1D advection equation. Our test set is the average over 20 data points drawn from the same distribution as the training set. We see that the average error of the machine learned solver grows without bound because some fraction of the datapoints blow up as $t \to \infty$, while the globally stabilized and flux-limited machine learned solvers have bounded error as $t \to \infty$.

# E    Energy-Conserving Global Stabilization

Consider the Boltzmann equation $\frac{\partial f}{\partial t} + \frac{\boldsymbol{p}}{m} \cdot \boldsymbol{\nabla} f = \left(\frac{\partial f}{\partial t}\right)_{\text{coll}}$ from kinetic physics which describes the evolution of the particle distribution function $f$ in phase space $(\boldsymbol{x}, \boldsymbol{p})$ due to collisions with other particles [5]. Because $f$ conserves particles $\int f \, d\boldsymbol{x} \, d\boldsymbol{p}$, momentum $\int f\boldsymbol{p} \, d\boldsymbol{x} \, d\boldsymbol{p}$, and energy $\frac{1}{2m} \int f \, \boldsymbol{p}^2 \, d\boldsymbol{x} \, d\boldsymbol{p}$ while maintaining $f \geq 0$ and increasing the entropy $-\int f \log f \, d\boldsymbol{x} \, d\boldsymbol{p}$, then as $t \to \infty$ $f$ must evolve towards a Gaussian distribution. Yet if global stabilization were applied naively to a Boltzmann equation solver without preserving the right combination of these invariants, then $f$ could evolve towards a flat distribution function or some other physically incorrect state. Thus, while global stabilization may be useful, it is not by itself always going to be sufficient to ensure that the solution evolves to the correct state as $t \to \infty$.

To demonstrate how additional conservation laws might be enforced, we again consider the incompressible euler equations in eq. (8) but now attempt to enforce an additional conservation law: conservation of energy. Recall that in fig. 2b, energy was not conserved by any of the schemes considered. Energy will be conserved if $\int \boldsymbol{u} \cdot \frac{\partial \boldsymbol{u}}{\partial t} \, dx \, dy = \int \psi \frac{\partial \chi}{\partial t} \, dx \, dy = \sum_{i,j} \bar{\psi}_{i,j} N_{i,j} \Delta x \Delta y = 0$ and is expressed most simply as $\langle \bar{\psi}_{i,j} | \boldsymbol{N}_{i,j} \rangle = 0$ where $\bar{\psi}_{i,j}$ is the cell average of $\psi$ and $N_{i,j}$ is the time-derivative in the $i$th, $j$th cell. Conservation of mass, conservation of energy, and $\ell_2$-stability will therefore all be guaranteed for eq. (8) if the following transformation is applied to $\boldsymbol{N}_{i,j}$:

$$\boldsymbol{U}_{i,j} = \boldsymbol{\chi}_{i,j} - \langle \boldsymbol{\chi}_{i,j} \rangle \qquad \boldsymbol{M}_{i,j} = \boldsymbol{N}_{i,j} - \langle \boldsymbol{N}_{i,j} \rangle \qquad \bar{\boldsymbol{\phi}}_{i,j} = \bar{\boldsymbol{\psi}}_{i,j} - \langle \bar{\boldsymbol{\psi}}_{i,j} \rangle$$

$$\boldsymbol{W}_{i,j} = \boldsymbol{U}_{i,j} - \frac{\langle \boldsymbol{U}_{i,j} | \bar{\boldsymbol{\phi}}_{i,j} \rangle}{\langle \bar{\boldsymbol{\phi}}_{i,j} | \bar{\boldsymbol{\phi}}_{i,j} \rangle} \bar{\boldsymbol{\phi}}_{i,j} \qquad \boldsymbol{P}_{i,j} = \boldsymbol{M}_{i,j} - \frac{\langle \boldsymbol{M}_{i,j} | \bar{\boldsymbol{\phi}}_{i,j} \rangle}{\langle \bar{\boldsymbol{\phi}}_{i,j} | \bar{\boldsymbol{\phi}}_{i,j} \rangle} \bar{\boldsymbol{\phi}}_{i,j}$$

$$\boldsymbol{N}_{i,j} \Rightarrow \boldsymbol{P}_{i,j} + \frac{d\ell_2^{\text{new}}/dt}{\langle \boldsymbol{W}_{i,j} | \boldsymbol{G}(\boldsymbol{\chi}_{i,j}) \rangle} \boldsymbol{G}(\boldsymbol{\chi}_{i,j}) - \frac{\langle \boldsymbol{W}_{i,j} | \boldsymbol{P}_{i,j} \rangle}{\langle \boldsymbol{W}_{i,j} | \boldsymbol{W}_{i,j} \rangle} \boldsymbol{W}_{i,j} \tag{12}$$

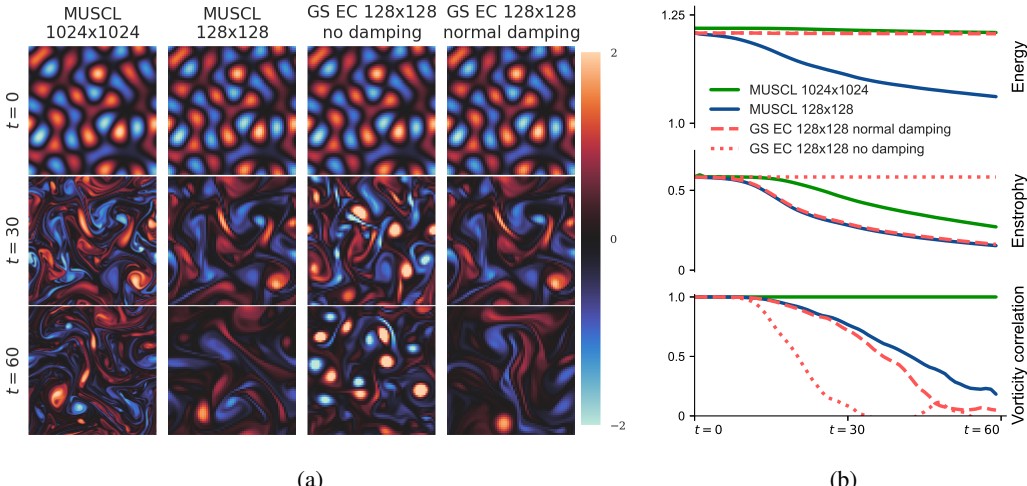

(a)                                                       (b)

Figure 5: Global stabilization can be modified to enforce energy conservation as well as stability for the incompressible Euler equations. (a) Images of the vorticity $\chi$ at three different times. The first and second columns show the baseline MUSCL scheme at high and low resolution. The third and fourth columns show the baseline MUSCL scheme with energy conserving global stabilization (GS EC), either with no numerical damping or with the normal rate of damping. (b) Energy, enstrophy, and vorticity correlation over time.

663    for any $d\ell_2^{\text{new}}/dt \leq 0$ and any non-constant scalar function $\boldsymbol{G}_{i,j}(\boldsymbol{\chi}_{i,j})$ for which $\langle \boldsymbol{G}_{i,j}(\boldsymbol{\chi}_{i,j}) \rangle = 0$,
664    $\langle \boldsymbol{G}_{i,j}(\boldsymbol{\chi}_{i,j}) | \bar{\boldsymbol{\psi}}_{i,j} \rangle = 0$ and $\langle \boldsymbol{W}_{i,j} | \boldsymbol{G}_{i,j}(\boldsymbol{\chi}_{i,j}) \rangle \neq 0$. A simple choice is $\boldsymbol{G}_{i,j}(\boldsymbol{\chi}_{i,j}) = \boldsymbol{W}_{i,j}$.

665    In fig. 5, we examine how the energy conserving global stabilization (GS EC) scheme in eq. (12)
666    affects the baseline MUSCL scheme. The third column and fourth columns of fig. 5a set $d\ell_2^{\text{new}}/dt = 0$
667    (no damping) and $d\ell_2^{\text{new}}/dt = \langle \boldsymbol{\chi}_{i,j} | \boldsymbol{P}_{i,j} \rangle$ (normal damping). As we can see in figs. 5a and 5b, the
668    energy-conserving schemes do conserve energy as predicted. Thus, for some equations it may be
669    possible to develop global stabilization schemes that enforce additional conservation laws.

