# OpenReview forum: "How To Design Stable Machine Learned Solvers For Scalar Hyperbolic PDEs"
_NeurIPS.cc/2022/Conference — NeurIPS 2022 Submitted_

### Official Review · Reviewer_sQzi · 2022-06-21

**Rating:** 3
**Confidence:** 4
**Soundness:** 4 excellent
**Presentation:** 2 fair
**Contribution:** 3 good

**Summary:**

I thank the authors for an interesting read.

Rollout stability of PDE solvers for time-dependent hyperbolic PDEs is a property baked into classical solvers, yet it is unaddressed in the machine learning for PDEs community. As such it is a common problem for learned solvers that they are sometimes unstable. When and under what conditions this instability arises is poorly understood and often swept under the table.

This submission seeks to resolve that with a technique that the authors call `global stabilization’. They introduce an L2-stability criterion, which states that the L2 norm of the solution is non-increasing over time. This is a well-known property of hyperbolic PDEs in conservation form. The beauty of this is that enforcing this property directly ensures that stability is met, since stability is defined as the solution blowing up and this cannot happen if solutions are norm non-increasing in time.

The technique is general and works as an ad hoc projection step to be used on the output of any model, but it is also handcrafted and there is no guarantee that it is the most general such stabilization method.


**Questions:**

I think a short discussion on different kinds of stability would be helpful. The kind focussed on in this submission is “eigenvalue stability” (aka weak stability, absolute stability, time-stability, practical stability, or P-stability) where the timestep is kept finite and time is allowed to tend to infinity as opposed to simply “stability” (aka zero-stability, D-stability, Lax stability, or Lax-Richtmyer stability) where time is kept fixed and the timestep tends to zero.

Conditions 2 and 3 on page 2 state that mass conservation and discrete L2-norm non-increase require the continuous time limit. Is this strictly true? One could discretize time again and these (in)equalities still hold.

While the four conditions on page 2 are sufficient, they do not guarantee accuracy (and therefore convergence). (After reading through the end I note here that this point is acknowledged in the limitations section). What is missing to guarantee accuracy?

Equation 5 is just one way to satisfy the L2-stability condition, but it does not look to be the most general expression that could be used here. Why did you choose this expression? Am I perhaps wrong and is this the most general expression that could be used?

Figure 1: it is not clear from the figure that at t=0.5 the red solution is less stable than the blue solution.

Lines 158 - 160: I see that there is a choice to be made here as to whether the ML solver should model the properties of the continuous or the discrete equations. To me it does not necessarily have to be one or the other. Is there a natural reason why you assert it must be that the ML solver must inherit the properties of the discrete equations only?

Is it such a limitation that global stabilization breaks causality? In many numerical systems it often happens that discretization breaks notions of locality. For instance, computing high order derivatives at a point requires that we use very large stencils, despite the fact that derivatives are defined entirely locally.


**Limitations:**

I believe that the authors have addressed most of the limitations of this work. The one remaining one is that I do not believe the proposed technique to be the most general possible solution and as such it may be that ML solvers will lose functional expressivity on some level.

**Strengths And Weaknesses:**

The paper tackles a very important and much overlooked issue in the ML for PDE solving literature. I welcome this contribution and see it as a positive aspect of the submission.

The paper boasts a nice related work section, which situates in the landscape of contemporary ideas adequately.

The limitations section of the paper is very open and honest. Furthermore it opens a lot of interesting questions, such as, how do we make accurate ML solvers? This makes it particularly transparent for a machine learning paper. I thank the authors for this addition.

The technique is general and works as an ad hoc projection step to be used on the output of any model, but it is also handcrafted and there is no guarantee that it is the most general such stabilization method. As such, as the authors admit themselves, while the method achieves the desired stability, it does not guarantee accuracy and looking at their results in Figure 3, it is certainly the case that standard measures of performance such as vorticity correlation take a large hit. This is the greatest weakness of the paper.

From what I can make out, the paper has no explicit experimentation section. I think this makes it hard to verify its efficacy.

---

> ### Author Response · Authors · 2022-08-02
> **Response to reviewer sQzi**
>
> Dear reviewer sQzi,
>
> Thanks for a truly excellent review with very insightful questions. Your comment about “the most general possible solution” is particularly insightful and helpful.
>
> We have submitted a revised paper. The revised paper has a more clear and rigorous presentation, with significant changes to sections 2, 3, and 6. We believe that the revised paper addresses all of your concerns, except your concern that "the paper has no explicit experimentation section." We will update the paper by August 9th.
>
>
> We agree with almost all of your comments. The only comment we think is incorrect is the comment “looking at their results in figure 3, it is certainly the case that standard measures of performance such as vorticity correlation take a large hit. This is the greatest weakness of the paper”. Sorry if this was confusing. Figure 3 is not related to the main discussion, we have moved it to an appendix in the revised paper. The proper conclusion to draw from figure 3 is instead that “it is often not even desirable for a numerical method to inherit the properties of the continuous equation” (see section 9 of revised paper).
>
> However, your comment (implicitly) raises an important question raised also by reviewer 9qVo: how will global stabilization influence the accuracy of a machine learned numerical method? The revised paper is more clear on this question (see sections 6.1 and 9). We have a theoretical justification for our argument in sections 6.1 and 9, but we agree it would be easier "to verify its efficacy" if we also had an experimental demonstration of our claim. We will work on updating the paper with experiments by August 9th.
>
>
> Questions:
>
> Q1: Excellent suggestion. We have rewritten sections 2 and 3 to more rigorously and clearly define the concept of ‘stability’.
>
> Q2: You are correct, this is not strictly true. However, global stabilization cannot be applied to in the discrete time case. The problem is this introduces a quadratic term proportional to $(dt)^2$ (see $\langle N | N\rangle$ term in line 291 in the revised paper) which I don't know how to cancel via transformation of $N$.
>
> Q3: Nothing can guarantee perfect accuracy, except by using a convergent method and taking the limit $dx \rightarrow 0$. Fundamentally, the issue is that you have an infinite degree-of-freedom system which you are solving using a finite number of degrees of freedom. Some information will be lost, so some accuracy will be lost. Classical numerical methods typically get a sufficiently accurate solution by using (a) a convergent numerical method, (b) some kind of hand-crafted interpolation function, and (c) a sufficiently small grid spacing $dx$. Hybrid machine learned numerical methods try to outperform standard numerical methods by replacing (either explicitly or implicitly) the hand-crafted interpolation function with some sort of statistical interpolation function and using large $dx$. Thus, for a machine learned numerical method, accuracy comes down to performing accurate statistical inference (i.e., regression) under uncertainty. We attempt to clarify by adding the phrase "which consistently make accurate predictions about the time evolution of the solution" to line 331.
>
> Q4: The reason you would choose this expression is because it has a simple physical interpretation, but as you point out it is not the most general expression. See sections 4 and 6 of the revised paper.
>
> Q5: The red solution has blown up and the magnitude is too large to be seen in figure 1 at $t=0.5$. We modify the caption, and explicitly state that the red solution blows up while the discrete $\ell_2$-norm of the blue solution is conserved.
>
> Q6: This is a tricky question. The answer in the text is "because machine learned PDE solvers solve discrete equations that are designed to approximate $\chi^{\textnormal{exact}}_{i,j}$". To expand on this a bit further: we have a loss function to minimize as well as hard constraints which we force our algorithm to satisfy. If our hard constraints are incompatible with the result that minimizes our loss function, then we are necessarily going to make errors at each timestep. For time-dependent PDEs, even small errors can accumulate into big errors. The exception would be if somehow the errors cancelled out, but for all the properties we've looked at the errors don't cancel.
>
> Q7: Interestingly, reviewer 9qVo has almost the opposite concern, they see this limitation as “potentially problematic”. We are inclined to agree with you: we don’t think this is a very concerning limitation. In the revised paper, we have removed this limitation from section 8. We discuss this further in section 6.1. We can recognize that this is not a limitation when we see that global stabilization simply is adding a constant-coefficient diffusion term to the solution (see section 4), where the diffusion coefficient depends on the solution over the entire domain. Adding numerical diffusion when it is needed is not a limitation.

---

### Official Review · Reviewer_6knQ · 2022-07-11

**Rating:** 7
**Confidence:** 3
**Soundness:** 4 excellent
**Presentation:** 2 fair
**Contribution:** 4 excellent

**Summary:**

This paper studies the scalar hyperbolic PDE problems on a uniform grid with periodical boundary conditions. The paper proposes a numerical finite-volume solver that guarantees to generate stable solutions to the PDEs by construction. The key observation is to notice that the stability of the solution can be enforced by properly reconstructing the cell boundary flux so that it ensures a non-increasing L2 norm of the solution. Therefore, it proposes a correction scheme for the cell boundary flux called “global stabilization”. While the main result is theoretical, one can properly combine it with machine learning solvers to generate a stable solver.

**Questions:**

1. Line 71: Is the last equation = 0 because of the periodic boundary conditions (so that f_0 and f_last are known to be the same)?

2. Although I can see that Eqn. 5 guarantees the L2 stability, I cannot see how it is derived based on the explanation in lines 95-96. I’d appreciate it if more intuition can be provided when deriving Eqn. 5.

3. On a related note, lines 109-110 mentioned Eqn. 5 can be used for both stabilizing a solver and reducing numerical damping and presented two numerical experiments to support them. Is there any theoretical analysis on the numerical diffusion introduced by Eqn. 5?

4. Because of 2 and 3, I have found Sec. 5 a bit difficult to follow. Maybe a better way is to first state the goal is to come up with a correction scheme on f so that it stabilizes the solver and reduces numerical damping, then explains how this goal leads to a derivation of Eqn. 5.


**Limitations:**

I really like the fact that the authors are upfront about the limitations of the proposed approach, especially by pointing out many assumptions that they made about the PDE system and numerical schemes. To me, these issues (assuming periodical boundary conditions, regular grids, limited PDE types, etc.) are not deal-breakers but rather show the potential of inspiring follow-up papers.

**Strengths And Weaknesses:**

Strengths:
- The paper studies a very critical question that, in my opinion, is a bottleneck of many existing machine learning solvers for PDEs.
- The mathematical insight behind the proposed solution is convincing.
- I also appreciate that the proposed numerical scheme stabilizes the solution and removes numerical diffusion at the same time.

Weaknesses:
- The paper requires advanced background knowledge of PDEs and is not very friendly to readers unfamiliar with numerical methods for PDEs. I think having more illustrations will be nice, e.g., an inset around Eqn 3 that displays a uniform grid with locations of each variable u and f.

---

> ### Author Response · Authors · 2022-08-02
> **Response to reviewer 6knQ**
>
> Dear reviewer 6knQ,
>
> Thanks for the very helpful review. We agree with all of your comments and your suggestions for improving the paper.
>
> We have submitted a revised paper. I suspect you will find that the revised paper is more rigorous, easier to follow, and friendlier to readers who do not have advanced background knowledge of PDEs and ML-based solvers. I hope you will agree with us that the presentation of the revised paper is now much better.
>
> If you feel inclined, you might read the comments of the other reviewers, who have all raised intelligent and important concerns. I hope you will agree with us that the revised paper satisfactorily addresses these concerns. Nevertheless, we agree with reviewer sQzi and reviewer 9qVo that our paper would benefit from an experimental verification of the claim that global stabilization "can be used to stabilize machine learned PDE solvers without degrading the accuracy of an already-accurate solver." We will add an experimental verification by August 9th.
>
>
> Questions:
>
> Good questions and suggestions. We have updated the revised paper accordingly.
>
> "Is there any theoretical analysis on the numerical diffusion introduced by Eqn. 5?" Thanks for asking this question. Setting $\beta=1$ is equivalent to adding a spatially constant diffusion coefficient to the solution (see section 4 of revised paper).

---

> > ### Comment · Reviewer_6knQ · 2022-08-09
> > **Response**
> >
> > Thank you for the response and the careful revision. I have read all of them as well as the other reviews, and I plan to keep my score unchanged.

---

### Official Review · Reviewer_9qVo · 2022-07-12

**Rating:** 3
**Confidence:** 4
**Soundness:** 2 fair
**Presentation:** 2 fair
**Contribution:** 2 fair

**Summary:**

The paper is concerned with designing stable solvers for scalar conservation laws on rectangular domains with periodic boundary conditions.
The central idea is to translate a stability condition into the requirement that in the continuous-time limit, the discrete $L^2$-norm of the solution must be non-decreasing.
This is equivalent to requiring $\sum_j u_j(f_{j+1/2} - f_{j-1/2}) \geq 0$ ($u_j$ and $f_{j\pm1/2}$ are discretisations of the variables in the PDE).
Now the main idea of the paper is to rewrite this inequality condition with summation-by-parts and to identify what needs to be added to each $f_{j+1/2}$ for the condition to always hold.



**Questions:**


* Line 25: "and as a result, often do not maintain stable solutions". I would appreciate a reference that reveals/discusses this issue.
* Line 31: "the answer to this question" Which question?
* Lines 24/25 and 27-34: I would appreciate a more formal definition of stability early in the paper. I find the exposition in 27-34 too high-level to be able to falsify/verify its statements.
* Lines 40f: "We note that the method can also be used when the right hand side (RHS) of eq. (1) is nonzero by treating the right hand side terms using classical techniques and applying global stabilization only to the divergence terms": please elaborate. If necessary, in an appendix. The statement, taken as is, is difficult to falsify/verify.
* Line 52: what is $S$? Please define all terms in the equation.
* Line 54f: "we can see that the discrete L2 norm is non-increasing": Please define these terms more rigorously; the paper only states $d/dt \int u^2/2 dx \leq 0$.
* Lines 54f: "we now list four conditions sufficient for ensuring that a solver of eq. (1) is stable:" Please embed the four conditions that you list into the literature. Does everyone mean these four conditions when discussing stability? Some references would help the understanding.
* Line 57: "the integral of the solution (...) is conserved": this statement would be easier to follow if you wrote down the integral, because it is not obvious whether the integral is over time, space, or both.
* Line 65: "Integrating eq. (1) over a grid cell" Please elaborate/formalise. For someone that does not already know finite volume schemes, I expect that it would be difficult to follow.
* Line 67: Please define $f_{j+1/2}$ prior to Equation (3).
* Lines 79/80: Please elaborate on the terms "spurious oscillations" and "numerical diffusion".
* Line 96f: "Thus, eg. (4) will be satisfied if the following transformation is made" Could you please write down the proof of this statement somewhere? This might be the most central contribution of the paper, it should not be up to the reader to verify it.
* Eq. (5): what does the $\equiv$ symbol mean? Please define it.
* Line 108: "which violates the physical property..." The paper mentions this in Section 8, and I also find it potentially problematic if each $f_{j + 1/2}$ depends on the solution over the entire domain because it seems to differ quite drastically from conventional method-of-lines- or finite-volumes-solvers.
I would appreciate a more thorough discussion of this possible drawback.
* Figure 1: I am not able to deduce from the figure how "global stabilization ensures that the discrete L2 norm is conserved."
* Figure 2(b): the two different shades of dark blue are incredibly difficult to distinguish.
* Lines 168f: "that adjusts the (...) output if that output moves the solution towards instability": what is the price in accuracy? I find it difficult to believe that adjusting for stability does not cost accuracy. This aligns with the statements made in Lines 255f, but is there any way of understanding or at least quantifying this trade-off?
* Eq. (10): please define the notation $\langle N \mid 1 \rangle$. I suppose it refers to an inner product?


**Limitations:**

In my view, the biggest limitation of the method seems to be understanding how much accuracy, respectively general reliability of the solver is traded for improved stability.  This is mentioned in Section 8, but not discussed very thoroughly (but other, more minor limitations are discussed thoroughly).


**Strengths And Weaknesses:**

I appreciate the idea presented in this work, and I like the honest list of the limitations of the proposed method.
But I am unsure about the motivation/premise behind the outlined algorithms, I struggle with the lack of mathematical rigour, especially in the introductory Sections 2 and 3, and I have too many questions about the proposed scheme to be able to recommend acceptance.
I think this paper should be rejected.
More details on what I consider strengths and weaknesses are below.

### Strengths

* Machine-learned PDE solvers are a subject that seems to be of interest to the Neurips community
* Designing a stable finite volume solver is an important task for PDE simulation
* The general idea is simple and could be implemented by anyone that wishes to adopt the proposed method
* Related work and limitations are assessed thoroughly


### Weaknesses

* The paper claims to be concerned with _machine-learned solvers_, but the content of the paper is independent of _learning_ solvers: the results apply to any algorithm that predicts the flux $f_{j + 1/2}$.
The two experiments do not learn PDE solvers but consider a-priori-chosen schemes: centred flux, and the _already-stable_ MUSCL scheme.
I am not sure whether this lack of a connection to machine learning makes the results less or more interesting. But given the venue and the paper's title and abstract, the disparity between mentioning "machine-learned PDE solvers" frequently on the one hand, and not connecting to learning PDE solvers, on the other hand, makes the message of the paper slightly confusing.
* The motivation for this paper is stated in line 88f.: "While machine-learned FV solvers are capable of ensuring stability via classical techniques [27, 25], we anticipate that such choices of flux will be too restrictive and too diffusive to outperform state-of-the-art numerical methods."
It seems like the submitted paper provides a solution to an _anticipated_ problem. There is no evidence, experimental or referenced, that shows how "such choices of flux will be too restrictive and too diffusive" is an actual concern.
In my view, this limits the expected impact of this work significantly.
* The readability of the paper is inhibited by a lack of rigour in technical components, especially in Sections 2 and 3. I understand that the authors attempt to keep the mathematical load on the reader as light as possible, but it makes the introduction difficult to follow, sometimes even difficult to verify/falsify. Some issues are listed in the questions below.

---

> ### Author Response · Authors · 2022-08-02
> **Response to reviewer 9qVo**
>
> Dear reviewer 9qVo,
>
> Thanks for a truly excellent and extremely thorough review. We agree with all your comments.
>
> We have submitted a revised paper. The revised paper has a more clear and rigorous presentation, with especially large changes to sections 2, 3, and 6. We believe that the revised paper addresses all of your concerns, although we do not define 'spurious oscillations' or 'numerical diffusion'. We will also update the paper by August 9th to add an experimental section.
>
>
> Limitations:
>
> "the biggest limitation of the method seems to be understanding how much accuracy... is traded for improved stability." We agree that this point is crucial but poorly explained. The revised paper is more clear on this topic (see sections 6.1 and 9).
>
> Although we have a theoretical argument for why global stabilization "can be used to stabilize machine learned solvers without degrading the accuracy of an already-accurate solver", we think that the paper would be more convincing if it had an explicit experimental demonstration of this claim. We have not had time to perform these experiments yet, but we will work on updating the paper with these experiments by August 9th.
>
>
> Questions:
>
> Most of your questions have been clarified in the revised paper. Some questions require further explanation:
>
> Lines 79/80: We have not defined these terms in the revised paper, but have cited a reference about numerical diffusion. We can define these terms in the August 9th resubmission if you still would like us to. The issue is that "spurious oscillations" and "numerical diffusion" are both elementary and fundamental concepts in numerical analysis, but neither concept has a rigorous definition or a simple-to-digest explanation. Spurious oscillations relate to a violation of the monotonicity property of the solution, which would require recursive explanation. Likewise, numerical diffusion is a topic that shows up again and again in numerical analysis but does not admit a simple explanation. A simple but non-rigorous explanation is that numerical diffusion is (explicit or implicit) diffusion added to a high-order method. Usually, numerical diffusion is used to ensure that a stability property is preserved.
>
> Line 108: Interestingly, reviewer sQzi has almost the opposite concern, they ask "is it such a limitation"? We are inclined to agree with reviewer sQzi: we don’t think this is a very concerning limitation. In the revised paper, we have removed this limitation from section 8. We discuss this further in section 6.1. We can recognize that this is not a limitation when we see that global stabilization simply is adding a constant-coefficient diffusion term to the solution (see section 4), where the diffusion coefficient depends on the solution over the entire domain. Adding numerical diffusion when it is needed is not a limitation.
>
> Figure 2(b): We will fix this in the August 9th resubmission by changing the dark blue solution to green.

---

> > ### Comment · Reviewer_9qVo · 2022-08-03
> > **Response**
> >
> > Thank you for your clarifications. I appreciate the improved presentation in the most recent upload, which resolves a sizeable subset of my specific questions.
> > * I like the new Section 6.1, but I do not think it elaborates on how much accuracy is traded for stability. In the rebuttal, you mention a theoretical argument for stabilising machine learned solvers without degrading accuracy; it would be great if you could point me to the section/equation in the paper where such a guarantee can be found. It sounds like a powerful statement, but I cannot seem to find it. What am I missing?
> > * Lines 79/80: I think it would be a good idea to explain numerical diffusion and spurious oscillations in the paper, because not every reader will be familiar with those terms. (In my view, the high-level explanation you used in the rebuttal together with a reference would be plenty.) But ultimately, it is of course the authors' decision what to discuss in the paper.
> > * Line 108: Thanks for your reply. In my review, I did not call this a limitation but a _potential_ limitation which might warrant more discussion. I kind of agree with you and reviewer sQzi that there are examples which also seem to break causality (even though I disagree with the example sQzi gives -- I would still consider a large stencil to be a local phenomenon). Nevertheless, it is not extremely common, especially in the sense that with global stabilisation, a formerly "causality-preserving" method would not be "causality-preserving" any more. Therefore, I think a more thorough discussion of this fact would improve the presentation of the paper.

---

> > > ### Author Response · Authors · 2022-08-05
> > > **Response**
> > >
> > > Thanks for your suggestions on how to continue to improve the paper.
> > >
> > > --Lines 79/80: We will include a similar high-level comment in the August 9th revision.
> > >
> > > --Line 108: We see now that we mischaracterized your statement. Sorry about that. We agree that a large stencil should be classified as a "local" phenomenon, and with your point that a more thorough discussion of this fact would improve the presentation. We will include this in the August 9th revision.
> > >
> > > --"It would be great if you could point me to the section/equation in the paper where such a guarantee can be found": The relevant section is lines 355-362 in section 9.
> > >
> > > Our intention was to use section 6.1 to explain that while other methods for generating stability add a large amount of numerical diffusion, global stabilization adds the minimum amount of numerical diffusion necessary to stabilize the solution. We also wanted to point out that this is only possible if we perform a global calculation of the L2-norm, which explains why we chose to allow the solution to violate causality.
> > >
> > > Our intention was then to use lines 355-362 to complete the argument:
> > > (A) Global stabilization is only added if the algorithm violates the non-increasing L2-norm property.
> > > (B) A highly accurate ML solver is unlikely to frequently violate this property within the training distribution.
> > > (C) Even if it does violate the property, the additional numerical diffusion is the minimum necessary to correct the violation.
> > > Thus, "we can expect the effects of global stabilization to be infrequent, small, and applied only when necessary." (A) is in algorithm 1. (B) is justified in appendix B. (C) is discussed in section 6.1.
> > >
> > > Now, if you are a very astute observer (and it seems like you are), you will point out that you it is possible to think of examples of PDEs and initial conditions where (B) is not true. Thus, we should do a better job of spelling out exactly when (B) is going to be true and when it might be violated.
> > >
> > > The important things for the reader to understand are that:
> > > (1) We assume that the training data is given by the coarse-grained exact solution.
> > > (2) For non-linear f(u) the solution develops high-$k$ structures or modes which coarse graining integrates out, meaning that the training data will have non-increasing discrete L2-norm with very high probability.
> > > (3) For linear $f(u)$, i.e., the advection equation, the solution doesn't develop high-$k$ structures, which means that for some initial conditions the discrete L2-norm of the training data will oscillate around a fixed value.
> > > In other words, only for linear f(u) and only for some initial conditions is (B) likely to be violated.
> > >
> > > We will move the theoretical argument to section 6.1, clarify that (B) is true for non-linear $f(u)$ but might not always be satisfied for linear $f(u)$, as well as point out that for linear equations it may be better to set $\frac{d\ell_2^{\textnormal{new}}}{dt}=0$ rather than $\frac{d\ell_2^{\textnormal{new}}}{dt}\le 0$.

---

### Official Review · Reviewer_ee6A · 2022-07-15

**Rating:** 3
**Confidence:** 4
**Soundness:** 2 fair
**Presentation:** 2 fair
**Contribution:** 2 fair

**Summary:**

The paper presents a set of sufficient criteria that guarantee the stability of numerical schemes used to solve scalar hyperbolic PDEs with periodic BCs and presents a predictor-corrector algorithm to implement such schemes. The prediction step is performed by a Machine Learned function (can also be other functions? such as Method of Lines) and finds the values of fluxes at desired locations to propagate the flow. The corrector step modifies the predicted flux values to ensure "sufficient criteria for stability" are satisfied.
The approach was applied to two examples, and the results show that this approach achieves "stabilization of unstable numerical schemes" and "reduction of numerical damping" while globally preserving conserved properties.

**Questions:**

A) Is there a way to alter C in equation (5) to be a function of index j and replace it with C_j in the corrector step to allow for local conservation of properties while preserving discrete L2-norm?

B) Equations 3b cannot be exact since flux at cell boundaries can still vary in a direction parallel to the cell boundary. What does ``exact'' mean in this context?

C) Why does the discrete energy of the system increase in the second numerical example in the globally stabilized scheme with no damping? What is the source of this energy coming into the system?

D) How do the 'beta'-parameters in the corrector step influence the performance/accuracy of the numerical scheme?

**Limitations:**

The application of results seems limited - there does not seem to be an explanation of how "sufficient conditions for stability" (see section 2) are derived, nor is there a guideline to obtain such criteria for other PDEs and boundary conditions. While the paper states this method could be extended for hyperbolic PDEs with a source term, higher spatial dimensions, and other boundary conditions but does not explain how. In that sense, the application is limited and specific to a particular PDE. Further, extension to vector-valued hyperbolic PDEs does not seem trivial as sufficient conditions are likely to change.


Soundness:
There are no apparent errors in the math; however, the main results of the paper are not compartmentalized into "theorems and proofs." Hence the exact limitations of the method are not explicitly stated anywhere. Furthermore, all the summation terms are missing index limits. Therefore actual treatment of the boundary terms and correction applied to fluxes at the boundary is not clear (the correction to fluxes is probably the same for interior and boundary  since periodic boundary conditions on an \R-interval are the same as no boundary conditions on a spherical manifold; however, such details are missing in the paper).

Presentation:
The paper's content is presented concisely and to the point; however, repeated use of technical terms without ever defining them leads to a lack of clarity. Many statements made are inadequately explained or supported. For example, the last sentence of the introduction, missing summation index limits in all the equations, last paragraph in section 3, second paragraph in section 5, etc.

Contribution:
Lack of novelty (a standard predictor-corrector approach), lack of mathematical rigor in the presentation, and limited application (only hyperbolic PDEs with periodic BCs, Method of lines, scalar PDE) make the overall contribution to be marginal.

**Strengths And Weaknesses:**

Strengths:

a) The algorithm (and the predictor-corrector approach) presented is relatively simple to implement in various numerical schemes

b) the results seem to indicate improvement in stability and numerical damping effects with almost no computational cost increase

Weakness:

a) Hard to generalize to other systems

b) Conserves properties (only?) on a global scale, and accuracy deteriorates in the presence of sharp gradients in the solution (for example, shocks in the flow).

---

> ### Author Response · Authors · 2022-08-02
> **Response to reviewer ee6a**
>
> Dear reviewer ee6a,
>
> Thanks for the helpful review. We have submitted a revised paper. The revised paper has a more clear and rigorous presentation. We believe that the revised paper addresses all of your concerns, except your concern that "there [is not] a guideline to obtain such criteria for [systems of hyperbolic] PDEs".
>
> You ask good questions and we agree with many of your comments. There are, however, a couple factual errors. As suggested by the Program Chairs, we will focus on the factual errors then answer your questions.
>
>
> Factual errors:
>
> 1) "Conserves properties (only?) on a global scale": In continuous systems, conservation is a global invariance that applies to closed systems. There is no such thing as 'local' conservation of, e.g., energy. Instead, you have continuity equations which describe the advection of, e.g., energy density. In discrete systems, conservation is also a global property, there is no such thing as 'local' conservation. For example, the discrete conservation of mass is derived by summing over all N grid cells (see lines 90-91 of revised paper). Instead, you have discrete continuity equations. Perhaps what you mean to say is that a discrete analogue of the entropy inequality is not satisfied for each 'local' grid cell but is satisfied globally. We do not think this is a weakness. We argue in section 7 that violating this condition is necessary to obtain an accurate solution.
>
> 2) "accuracy deteriorates in the presence of sharp gradients in the solution (for example, shocks in the flow)": Perhaps you are misinterpreting figure 1. Figure 1 demonstrates that if you apply global stabilization to a numerical method which is unstable and highly inaccurate, you will get a result which is stable and less inaccurate, but still inaccurate. As we make clear in section 4 of the revised paper, global stabilization adds diffusion to the solution. This will not degrade accuracy near shocks, in fact shock-capturing methods all add numerical diffusion near shocks.
>
> 3) "Equations 3b cannot be exact since flux at cell boundaries can still vary in a direction parallel to the cell boundary": Flux is perpendicular to a surface. The flux parallel to a surface is by definition zero. Equation 3b is exact; it is simply a discrete statement of the continuity equation for a rectangular cell.
>
> 4) "limited application": While you correctly state that the original paper "does not explain how" to extend the method to (a) higher dimensions, (b) non-periodic boundary conditions, and (c) systems of hyperbolic PDEs, you incorrectly conflate "not explain[ing] how" with having a "limited application". In other words, just because we do not explain how to extend the method to (a) (b) and (c) does not mean that our method cannot be extended to (a) (b) and (c). Students familiar with FV methods will be able to derive (a) by deriving a 3D analogue of the FV update equations 3a and 3b then computing the rate of change of the L2 norm. As we explain in the revised paper, the method can be extended to (b) by keeping track of the (known) fluxes through the domain boundary. The energy method can also be extended to some systems of hyperbolic PDEs, though we do not include "a guideline for obtaining such criteria". Nevertheless, as we argue in section 8 of the revised paper, "it is standard practice in the numerical analysis community to first use the scalar conservation law eq. (1) to introduce a new method before later extending the method to systems of PDEs". See, for example, Cockburn and Shu's papers introducing the Discontinuous Galerkin method.
>
>
> Questions:
>
> A) Good point. Reviewer sQzi has a similar question, stating that equation 5 "does not look to be most general expression". We have updated sections 4 and 5 to be more general.
>
> C) The discrete energy increases for the no damping case for the same reason the energy decreases in the MUSCL 128x128 case: numerical error. Although there is not a clear physical interpretation here, I think the best way to understand the increase in energy is that high-k oscillations in vorticity space lead to high-magnitude variations in velocity space, which lead to an increase in the energy.
>
> D) Good question. Setting beta=$ has "a simple physical interpretation: [it] correspond[s] to the addition of a spatially constant diffusion coefficient everywhere in space." See section 4 of the revised paper.
>
>
> Finally, you conclude that there is a 'lack of novelty' due to this being 'a standard predictor-corrector approach'. Even if you conclude that our technique is not novel, as stated in the reviewer guidelines a novel combination of well-known techniques can be valuable, especially when doing so makes progress on an important problem.

---

> ### Comment · Reviewer_ee6A · 2022-08-04
> **response to response to review of paper 3441**
>
> After reviewing the response, I do not believe the authors have addressed the substance of my more significant concerns.
>
> 	Factual errors:
> 	AR: A) "Conserves properties (only?) on a global scale": In continuous systems, conservation is a global invariance that applies to closed systems. There is no such thing as 'local' conservation of, e.g., energy. Instead, you have continuity equations which describe the advection of, e.g., energy density. In discrete systems, conservation is also a global property, there is no such thing as 'local' conservation. For example, the discrete conservation of mass is derived by summing over all N grid cells (see lines 90-91 of revised paper). Instead, you have discrete continuity equations. Perhaps what you mean to say is that a discrete analogue of the entropy inequality is not satisfied for each 'local' grid cell but is satisfied globally. We do not think this is a weakness. We argue in section 7 that violating this condition is necessary to obtain an accurate solution.
>
>
> Reviewer Response (RR): Yes, the issue is enforcement of continuity and entropy constraints on every grid.
>
>
> 	AR: B) "accuracy deteriorates in the presence of sharp gradients in the solution (for example, shocks in the flow)": Perhaps you are misinterpreting figure 1. Figure 1 demonstrates that if you apply global stabilization to a numerical method which is unstable and highly inaccurate, you will get a result which is stable and less inaccurate, but still inaccurate. As we make clear in section 4 of the revised paper, global stabilization adds diffusion to the solution. This will not degrade accuracy near shocks, in fact shock-capturing methods all add numerical diffusion near shocks.
>
> RR: Unless I misunderstand something, these high-frequency oscillations near the shocks, as seen in the figure 1, are unwanted and "unphysical". Hence, I described it as inaccurate because there are numerical schemes that do not have this problem. You could discuss whether an improvement can be made to eliminate these spurious oscillations, however, the current paper does not yet have that.
>
>
> 	AR: C)	"Equations 3b cannot be exact since flux at cell boundaries can still vary in a direction parallel to the cell boundary": Flux is perpendicular to a surface. The flux parallel to a surface is by definition zero. Equation 3b is exact; it is simply a discrete statement of the continuity equation for a rectangular cell.
>
> Perhaps the authors have not understood the issue. I meant to say that 'flux that is perpendicular to the face' can vary along the edge. For example, if you consider an edge [0,1] then the flux in the interval [0,0.5] can be pointing inwards, whereas the flux in the interval [0.5,1] can be pointing outwards. Regardless, the paper now states averaged flux which is fine i think.
>
> 	AR: D) "limited application": While you correctly state that the original paper "does not explain how" to extend the method to (a) higher dimensions, (b) non-periodic boundary conditions, and (c) systems of hyperbolic PDEs, you incorrectly conflate "not explain[ing] how" with having a "limited application". In other words, just because we do not explain how to extend the method to (a) (b) and (c) does not mean that our method cannot be extended to (a) (b) and (c). Students familiar with FV methods will be able to derive (a) by deriving a 3D analogue of the FV update equations 3a and 3b then computing the rate of change of the L2 norm. ............ Nevertheless, as we argue in section 8 of the revised paper, "it is standard practice in the numerical analysis community to first use the scalar conservation law eq. (1) to introduce a new method before later extending the method to systems of PDEs". See, for example, Cockburn and Shu's papers introducing the Discontinuous Galerkin method.
>
> RR: Perhaps 'Limited application' is not right choice of words; 'the application shown in the paper is limited to a specific pde' is more accurate. With that being said, unless some amount of explanation or proof is provided 'blanket statements' should be avoided. Assertions, made in the paper, should either be self-explanatory or supported by a proof/reference.
>
> 	AR: Finally, you conclude that there is a 'lack of novelty' due to this being 'a standard predictor-corrector approach'. Even if you conclude that our technique is not novel, as stated in the reviewer guidelines a novel combination of well-known techniques can be valuable, especially when doing so makes progress on an important problem.
>
> RR: 'Valuable' and 'novel' are not the same. The 'lack of novelty' remark was made because the paper claims the techniques presented as 'novel' which is not the case. It would be better to focus on the contributions made by the paper and the value they add.

---

> > ### Author Response · Authors · 2022-08-05
> > **Response to response to response to reviewer ee6a**
> >
> > A) Although our method doesn't maintain a discrete analogue of the entropy inequality in each grid cell, neither does the MUSCL scheme (reference [36], equation 3.16). Thus, we don't think this is a weakness of our scheme.
> >
> > B) It is not a factual error to say that the solution is inaccurate, but it is a factual error to say that "accuracy deteriorates" when in fact accuracy improves. Eliminating spurious oscillations from the unstable and inaccurate centered flux would require designing a TVD scheme. An accurate machine learned flux predictor would probably not introduce spurious oscillations.
> >
> > Regarding (A) and (B), we believe our paper would be substantially weaker if our method satisfied a discrete entropy inequality or was TVD. We allude to this in lines 348-354, writing that "designers of numerical methods must determine which properties of the continuous system should be preserved by the discrete system and which properties either cannot be preserved or degrade the accuracy of the discrete system."
> >
> > C) We misunderstood your comment. Sorry about that. Glad you agree the equation is in fact exact.
> >
> > D) We can include an appendix which derives the expressions for non-periodic boundary conditions and non-uniform grid spacing. We will remove the sentence "Our results can easily be generalized to the case where the grid spacing is non-uniform."
> >
> > E) It seems like you want us to remove the word "novel" from line 123, which we are happy to do.

---

### Author Response · Authors · 2022-08-02
**Submission of revised paper**

Reviewers,

Thank you for the excellent and detailed reviews. We are very impressed by the quality of the reviews.

As you can see, we have submitted a revised paper. We hope that you would please read the revised paper, especially sections 2, 3, 6, and 9. We hope you will agree with us that the updated paper has an improved presentation and that it addresses your concerns.

In our opinion, the remaining weakness of the revised paper is that we only have theoretical justification for our claim that global stabilization "can be used to stabilize machine learned PDE solvers without degrading the accuracy of an already-accurate solver." This claim would be more convincing if it also had experimental evidence to support it.

To address this remaining weakness, we plan to submit another revision of the paper by August 9th which tests our claim experimentally. We will run experiments using a machine learned solver for the 1D advection equation (plus a small diffusion term). The 1D advection equation is a very important model equation for fluid-like equations that is "embarrassing[ly]" difficult to solve accurately. We expect the solver to be quite accurate and stable within the training distribution, and for these experiments to show that global stabilization has only a small impact on the accuracy of the solver.

Thanks,
The Authors

---

### Author Response · Authors · 2022-08-09
**Second Revision of Paper**

Reviewers,

We have submitted a second revised paper.

Our paper now contains an experimental verification of the claim that global stabilization "can be used to stabilize machine learned PDE solvers without degrading the accuracy of an already-accurate solver." See lines 247-251 in section 6.1, figure 3, and appendix D.

Other revisions include:

--Clarifying the theoretical argument in section 6.1. See lines 234-246 and Appendix C.

--Defining 'spurious oscillations' and 'numerical diffusion'. See lines 106-108.

--Discuss the violation of finite propagation speed (i.e., causality). See lines 340-351.

--Generalize equation (12). Now vorticity correlation takes less of a hit in figure (5). See appendix E.

--Add an appendix generalizing to non-periodic boundary conditions and non-uniform grid spacing. See appendix B.

--Non-supported assertions have been removed.


Thanks, The Authors

---

### Meta-Review · Area_Chair_siAj · 2022-08-25

**Recommendation:** Reject
**Confidence:** Certain

**Metareview:**

Thank you for your submission to NeurIPS. All four reviewers authors are enthusiastic of this work, though three of the four reviewers had major concerns with the actual submission. In response, the authors carried out a major revision within a very short turn-around, completely rewriting most of the paper. All four reviewers are highly appreciative of the authors' responsiveness in replying to the reviews, in submitting new revisions taking into account reviewer comments.

Unfortunately, some concerns remain even after the author-reviewer discussion period closed. Broadly, the three reviewers all felt that there had been too many edits of the paper to be able to verify/falsify the newest content in all detail. The new revision is sufficiently different as to necessitate a fresh set of reviews, in a way that the conference format of NeurIPS is not able to facilitate.

Overall, the core idea is interesting and potentially very useful. The paper studies a critical problem typically overlooked by the current trend of combining ML with PDE solvers, and that it is transparent and upfront about its limitations. The paper asks a good question that can inspire important follow-up works. I highly encourage the authors to take reviewer comments into account, to expand into a complete, detailed experimental section and compare against other ML-based approaches, and to resubmit to an upcoming ML conference.


**Award:**

No

---

### Decision · Program_Chairs · 2022-09-14

Reject